# Identifying Dependent Components from Multi-domain Linear Mixtures

**Danru Xu** [1]   **Lauri Parkkonen** [2]   **Sara Magliacane** [1 3]   **Aapo Hyvärinen** [4]

## Abstract

We study a linear mixing model with dependent latent components, assuming multiple data domains. Most existing models assume that the components are independent or at least uncorrelated, in line with independent component analysis (ICA). Some recent work allows for dependent components, but then makes specific assumptions such as parametric forms of dependencies, multi-view settings, or interventions, or does not recover the individual components. In contrast, we consider a multi-domain setting in which domains differ through domain-specific scalings of the components, while the distribution of the underlying latent components is the same across domains. This approach can model data collected, for example, from different sensors measuring the same process, different laboratories conducting the same experiment, different experimental conditions, or different subjects that might differ in biological or physiological factors. We show that, under sufficient domain variability, latent variables and mixing functions can be identified from second-order statistics alone. We propose the **Mu**lti-**Do**main **Co**variance **M**atching (MuDo-CoM) algorithm that generalizes previous methods of joint diagonalization. MuDo-CoM is validated on simulated data and a real-world fMRI dataset.

## 1. Introduction

Learning meaningful latent representations from high-dimensional observations is important across many scientific fields, including neuroscience (Calhoun et al., 2001; Smith et al., 2013), genomics (Leek & Storey, 2007; Stegle et al., 2012), and climate science (Hannachi et al., 2007). This assumes the observations can be modeled as entangled (mixed) measurements of underlying latent variables, also referred to as components or sources. Typically, the scientific goal of these studies is to better understand the underlying structure of the data using these latent variables. Often, the latent variables are assumed statistically independent, which has led to the rich literature on independent component analysis (ICA) (Hyvärinen et al., 2001). An important question is how to create a latent variable model in which the components can be dependent, while they are still element-wise identifiable (i.e. they can be uniquely recovered from the measurements).

While the basic models assume the data is observed i.i.d., in many applications, data are collected across multiple domains, such as different subjects, sensors, laboratories, or experimental conditions. One concrete example is in functional magnetic resonance imaging (fMRI) data, where observed signals are commonly interpreted as mixtures of latent neural sources, each ideally describing a brain region or a functional network (Erhardt et al., 2011). Importantly, data distributions vary across domains, e.g. due to differences in biological and physiological factors, such as age or disease state, lead to noticeable changes in the distributions. Moreover, the fMRI measurements are typically collected across different laboratories or sensors, introducing additional variability due to device-specific calibration or preprocessing (Costafreda, 2009).

Previous research has proposed different ways to model multi-domain data using latent variables. Some of the earliest work considered a model akin to independent component analysis, and assumed the variances of the components change over domains or time segments (Pham & Cardoso, 2001). Recent work in causal representation learning also considers dependent components in a multi-environment setting. This typically requires latent variation across domains using techniques such as interventions (Lippe et al., 2022; 2023b; von Kügelgen et al., 2023; Ahuja et al., 2023; Sturma et al., 2023; Zhang et al., 2023; Ng et al., 2025), actions (Lippe et al., 2023a) or specific sparsity patterns (Lachapelle et al., 2024; Xu et al., 2024). However, in many environments, we might not have access to interventional data or actions, and the sparsity assumptions might not hold. A different approach is to consider the data as multi-view (Von Kügelgen et al., 2021; Richard et al., 2021; Daunhawer et al., 2023; Yao et al., 2024), which means that the datapoints in different domains are paired (e.g. measured

---

[1]University of Amsterdam, Netherlands [2]Aalto University, Finland [3]Saarland University, Germany [4]University of Helsinki, Finland. Correspondence to: Danru Xu <dxu3@uva.nl>.

*Proceedings of the 43rd International Conference on Machine Learning*, Seoul, South Korea. PMLR 306, 2026. Copyright 2026 by the author(s).

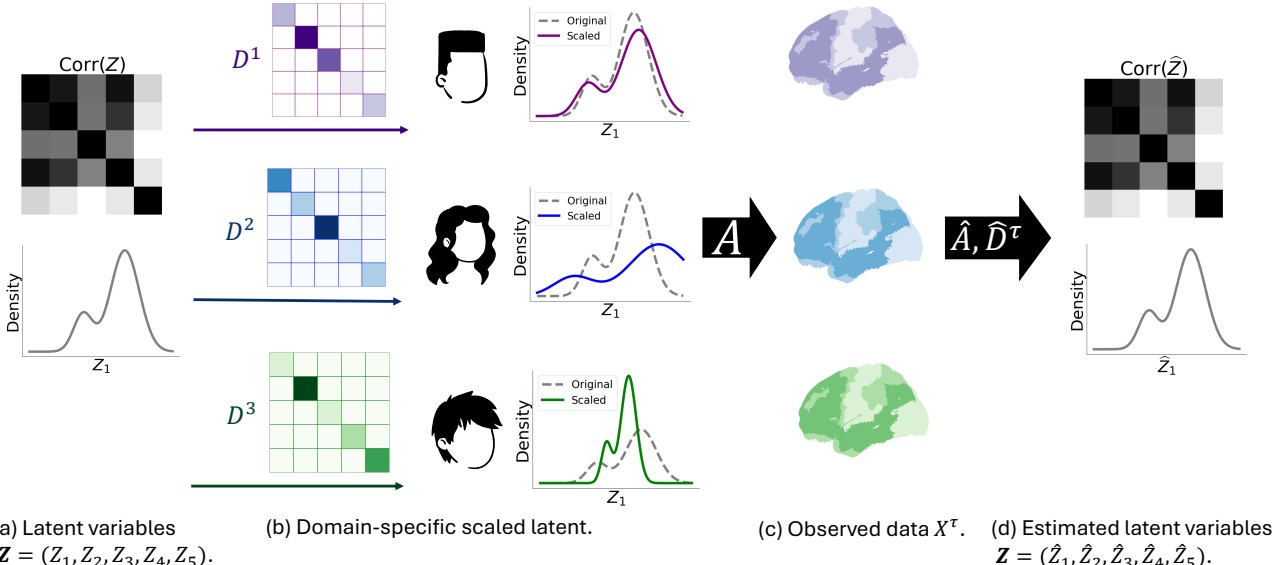

(a) Latent variables
$\mathbf{Z} = (Z_1, Z_2, Z_3, Z_4, Z_5)$.

(b) Domain-specific scaled latent.

(c) Observed data $X^\tau$.

(d) Estimated latent variables
$\mathbf{Z} = (\hat{Z}_1, \hat{Z}_2, \hat{Z}_3, \hat{Z}_4, \hat{Z}_5)$.

*Figure 1.* Motivating example based on fMRI data. *(a):* latent variables $\mathbf{Z} = (Z_1, \ldots, Z_5)$ with nontrivial dependencies, shown in their correlation matrix. *(b):* different domains (e.g., subjects) with their domain-specific diagonal scaling matrices $D^\tau$, which rescale the latent variables while preserving their dependency structure, as illustrated in the marginal distribution of $Z_1$. *(c):* the scaled latents are mixed through a shared, domain-invariant mixing matrix $A$ to generate observed data $X^\tau$. *(d):* from these observations $X^\tau$, the goal is to recover the latent variables $\mathbf{Z}$ (up to permutation and scaling), capturing both inter-regional dependence and inter-domain variability.

simultaneously or responding to the same stimulus). However, such pairing is not always feasible in practice, e.g., when we have measurements from different populations.

Here, we consider the fundamental linear mixing model, where the components undergo a linear transformation to produce the observed variables. In spite of decades of research, there is still no model that would allow for element-wise identifiability of the components, if they have *arbitrary dependencies* with *arbitrary linear mixing*, especially without any temporal dependencies or hard interventions. This is clearly of fundamental theoretical importance. Linearity in fMRI is a widely-held assumption, starting from linear regression models in the 1990s, and in resting-state studies, linear ICA is a standard method of analysis; early justification can be found in Buxton et al. (2004); Damoiseaux et al. (2006). On the other hand, neuroscientific evidence indeed shows strong *inter-regional correlations* during both resting state (Fox & Raichle, 2007) and during tasks (Ranganath et al., 2005; Hampson et al., 2006). In this paper, we show how the case of arbitrary dependencies can be solved through a judicious use of the multi-domain setting.

We model the variability over domains by assuming that while the different domains have the same underlying latent variables, measurements are generated through *domain-specific scaling transformations*. For example, in fMRI, the levels of activations of such regions are quite individual, especially in the resting state (Lee et al., 2023). This would mean that the magnitudes of latent neural compo-

nents change over domains, e.g., different subjects, even though their underlying functional connectivity remains similar. This is illustrated in Figure 1. We make the following contributions:

- We introduce very general identifiability results for linearly mixed latent-variable models in multi-domain settings. We do not assume independence, specific distributions, or specific forms of dependencies. We only require finite second-order moments and sufficient variability in domain-specific scalings. This generalizes ICA-style identifiability to settings with *arbitrary dependencies* between the latent variables with *multi-domain* observations.

- We propose a multi-domain *covariance-matching estimation algorithm* that generalizes classical second-order methods (e.g., joint diagonalization and three-way tensor decompositions) by allowing correlated latent variables.

## 2. Problem Setting

We consider a setting in which the data-generating process is a latent variable model with variables in the vector $\mathbf{Z} = (Z_1, ..., Z_n)$, which may have **arbitrary dependencies** with each other. $\mathbf{Z}$ takes values in $\mathcal{Z} \subseteq \mathbb{R}^n$, and we denote its covariance matrix as $G \in \mathbb{R}^{n \times n}$, which we assume is an arbitrary positive definite matrix with finite entries. Importantly, we introduce no further restriction on the distribution

of $\mathbf{Z}$, such as (non) Gaussianity, beyond the existence of the second-order moments.

In our setting, we do not observe $\mathbf{Z}$ directly, but we instead observe a set of datasets of entangled measurements across $T$ *domains*, which differ only in terms of the *scaling* of the components. For example, consider the setting in which we have fMRI data for different subjects, as in Fig. 1. Here, latent variables $\mathbf{Z}$ can be interpreted as activities of regions or functional networks, which we assume have the *same dependency structure* across subjects and a *shared spatial or functional mapping* from latent neural sources to observed fMRI measurements, but might differ in terms of scaling due to inter-individual variability. Formally, we assume a *multi-domain setting* in which the observations $\mathbf{X}^\tau \in \mathcal{X}^\tau \subseteq \mathbb{R}^d$ in each domain $\tau \in [T]$ are generated as

$$\mathbf{X}^\tau = AD^\tau \mathbf{Z}, \tag{1}$$

where the model has the following parameters: $A \in \mathbb{R}^{d \times n}$ is an unknown domain-invariant full column-rank mixing matrix and $D^\tau$ is a domain-specific diagonal scaling matrix, defined as $D^\tau = \mathrm{diag}(d_1^\tau, ..., d_n^\tau)$ with $d_i^\tau \in \mathbb{R}, \forall i \in [n]$ and $\tau \in [T]$. Throughout the paper, $X^\tau$ denotes the observed data matrix from domain $\tau$, while bold symbols (e.g., $\mathbf{X}^\tau$) denote the corresponding random variables.

This formulation captures the idea that domains differ only through domain-specific rescaling of latent variables, while the underlying latent variables and the mixing structure remain common. In our fMRI example, the domain-invariant mixing matrix $A$ represents the shared mapping from the neural sources to the measurements, while the domain-specific diagonal scaling matrices $D^\tau$ capture each subject's individual variability in magnitude or variance in latent neural states. Another example could be data collected in different hospitals, potentially on different patient populations, where measurement devices are calibrated differently.

We emphasize that our model is different from a multi-view setting, where multiple views of the same latent variable are jointly observed or paired together, e.g., different measurements of the same subject. In our setting, the observations in the domains are not paired in any way, which allows us to model also settings with different patient populations across different domains, e.g., hospitals. Each domain provides an independent dataset generated from the same latent distribution but with a different mixing process. This distinction is crucial, as identifiability needs to be built here without relying on pair-wise correspondence across domains.

**Identifiability definition.** Our goal is to recover the (potentially) dependent latent variables $\mathbf{Z}$ from the observations in the different domains, as well as the mixing matrix and the domain-specific scaling matrices. In general, as shown in previous studies (Comon, 1994; Cardoso, 1998; Hyvärinen

et al., 2001), latent variables cannot be fully recovered from observations: they are identifiable up to an equivalence class. In the basic ICA case, only permutation and scaling indeterminacies remain.

In our multi-domain setting, however, observations are generated through domain-specific mixing functions that differ through scaling matrices $D^\tau$. We therefore need to adapt the classic definition of *identifiability up to permutation and scaling* used in single domains (Comon, 1994; Khemakhem et al., 2020; Lachapelle et al., 2023; Xu et al., 2024), to fit our multi-domain setting with domain-specific scalings. In fact, two different cases will be used in our theory:

**Definition 2.1.** Given an observation $\mathbf{X}^\tau = \mathbf{f}^\tau(\mathbf{Z})$ and a learned representation vector $\mathbf{g}^\tau(\mathbf{X}^\tau)$, the ground-truth latent variables $\mathbf{Z}$ are identified by $\mathbf{g}^\tau(\mathbf{X})$ up to:

(i) *permutation, scaling and domain-specific sign*, if there exist a permutation matrix $P$, an invertible diagonal matrix $L$, and for each domain $\tau \in [T]$ a sign matrix $S^\tau$ (a diagonal matrix with entries in $\{-1, 1\}$), such that $\mathbf{g}^\tau(\mathbf{X}^\tau) = PLS^\tau \mathbf{Z}$ for all $\tau \in [T]$.

(ii) *permutation and scaling*, if additionally there is no sign discrepancy across domains, i.e., $\mathbf{g}^\tau(\mathbf{X}^\tau) = PL\mathbf{Z}$.

## 3. Identifiability of Latent Variables

In this section, we study the identifiability of latent variables in the multi-domain setting with domain-specific rescaling described in Sec. 2. Importantly, multiple domains give additional information only when observations are collected under sufficiently diverse measurement conditions. If the domain-specific scaling matrices are too similar or degenerate (e.g., some entries are zero), this will result in the same information as a single domain.

To rule out such ill-posed cases, we require sufficient diversity in how the latent variables are rescaled across domains. We formalize this as follows:

**Assumption 3.1.** [Sufficient variability of domain scalings] There exist $m = \binom{n+3}{4}$ domains $\tau_1, \ldots, \tau_m$, each with a domain-specific diagonal scaling matrix $D^\tau = \mathrm{diag}(d_1^\tau, \ldots, d_n^\tau)$, such that the matrix

$$\begin{pmatrix} \mathbf{d}_{(4)}^{\tau_1} & \mathbf{d}_{(3,1)}^{\tau_1} & \mathbf{d}_{(2,2)}^{\tau_1} & \mathbf{d}_{(2,1,1)}^{\tau_1} & \mathbf{d}_{(1,1,1,1)}^{\tau_1} \\ \mathbf{d}_{(4)}^{\tau_2} & \mathbf{d}_{(3,1)}^{\tau_2} & \mathbf{d}_{(2,2)}^{\tau_2} & \mathbf{d}_{(2,1,1)}^{\tau_2} & \mathbf{d}_{(1,1,1,1)}^{\tau_2} \\ \vdots & \vdots & \vdots & \vdots & \vdots \\ \mathbf{d}_{(4)}^{\tau_m} & \mathbf{d}_{(3,1)}^{\tau_m} & \mathbf{d}_{(2,2)}^{\tau_m} & \mathbf{d}_{(2,1,1)}^{\tau_m} & \mathbf{d}_{(1,1,1,1)}^{\tau_m} \end{pmatrix} \tag{2}$$

is full-row rank, composed of the following sub-matrices:

$$\mathbf{d}^{\tau}_{(4)} = \{(d^{\tau}_i)^4\}^n_{i=1},$$

$$\mathbf{d}^{\tau}_{(3,1)} = \{(d^{\tau}_i)^3 d^{\tau}_j\}_{i,j\in[n] \text{ s.t. } i\neq j},$$

$$\mathbf{d}^{\tau}_{(2,2)} = \{(d^{\tau}_i)^2(d^{\tau}_j)^2\}_{i,j\in[n], \text{ s.t. } i<j},$$

$$\mathbf{d}^{\tau}_{(2,1,1)} = \{(d^{\tau}_a)^2 d^{\tau}_b d^{\tau}_c\}_{a,b,c \in[n], \text{ s.t. all distinct}, b<c},$$

$$\mathbf{d}^{\tau}_{(1,1,1,1)} = \{d^{\tau}_a d^{\tau}_b d^{\tau}_c d^{\tau}_d\}_{a,b,c,d \in[n], \text{ s.t. } a<b<c<d}.$$

In addition, $d^{\tau_j}_i \neq 0$ for all $i \in [n]$ and $j \in [m]$.

The requirement of $m = \binom{n+3}{4}$ arises from a worst-case identifiability analysis and should be viewed as a sufficient but not necessary condition. In practice, since real-world scaling factors typically vary continuously and are far from adversarial, substantially fewer domains often suffice. We empirically verify this in Sec. 6 and provide a simple example in Ex. B.1. Joint Diagonalization (Belouchrani et al., 2002) also uses similar sufficient variability of domains to achieve identifiability, and it can be regarded as a special case of ours; a detailed discussion is in Sec. 4 and App. C.

Now we can give our main identifiability theorems. Identifying the latent variables $\mathbf{Z}$ is fundamentally related to identifying the underlying mixing process, which in our setting is parameterized by $A$ and $\{D^{\tau}\}^T_{\tau=1}$. However, in the multi-domain case, these are not quite equivalent, as we will see below. As a first step, we establish identifiability results for the domain-invariant mixing matrix $A$:

**Theorem 3.2.** *Consider the setting described in Sec. 2. Suppose Ass. 3.1 is satisfied. Then, the domain-invariant mixing matrix $A$ is identifiable up to permutation and scaling (including sign switches), i.e., $A$ is recovered up to right multiplication by a monomial matrix* [1].

The proof is given in App. B.2.1. Regarding identifiability of the components, the situation is complicated by the fact that the domains have different scalings and signs. The identifiability of the components is analyzed in the following theorem (proven in App. B.2.2):

**Theorem 3.3.** *We assume the same conditions as Thm. 3.2. Then, the domain-specific mixing $\{AD^{\tau}\}^T_{\tau=1}$ can be identified up to right multiplication by a (domain-independent) monomial matrix and a domain-related sign matrix $S^{\tau}$ for all $\tau \in [T]$, leading to the identifiability of $\mathbf{Z}$ up to permutation, scaling, and domain-specific sign (Def. 2.1.(i)).*

*Moreover, if all entries of $D^{\tau}$ share the same sign for every $\tau \in [T]$, then $\mathbf{Z}$ is identifiable up to permutation and scaling for all domains $\tau \in [T]$ (Def. 2.1.(ii)).*

This theorem says that variability across domains allows us to disentangle the latent variables from the observations and

estimate how they are rescaled across domains. However, the signs of the components cannot be recovered in general, i.e., the signs may still flip from one domain to another independently without affecting the observed data. This leads to identifiability of the latent variables up to permutation, scaling, and domain-specific sign (Def. 2.1.(i)). However, if we further assume that the scalings in each domain share a common sign, then these sign indeterminacies are removed, enabling identifiability up to the classical permutation and scaling (Def. 2.1.(ii)).

## 4. Multi-Domain Covariance Matching

Based on the multi-domain latent variable setting and the identifiability results in Sec. 3, we introduce the **Mu**lti-**Do**main **Co**variance **M**atching (MuDo-CoM) algorithm for recovering latent component from multi-domain observations, composed of the two following steps.

**Step 1: Covariance–matching estimation.** For a multi-domain dataset $\{X^{\tau}\}^T_{\tau=1}$, we estimate $\hat{A}$, $\{\hat{D}^{\tau}\}^T_{\tau=1}$, $\hat{G}$ by minimizing the domain-wise covariance-matching loss as follows:

$$\min_{\hat{A},\{\hat{D}^{\tau}\},\hat{G}} \sum^T_{\tau=1} \left\| \widehat{\text{Cov}}(X^{\tau}) - \hat{A}\hat{D}^{\tau}\hat{G}\hat{D}^{\tau}\hat{A}^T \right\|^2_F, \quad (3)$$

where $\|\cdot\|^2_F$ denotes the squared Frobenius norm, i.e., the sum of squared entries. $\widehat{\text{Cov}}(X^{\tau})$ denotes the sample covariance matrix computed from data $X^{\tau}$ in domain $\tau$.

In the implementation, we parameterize $\hat{G}$ using a structural equation model (SEM)-inspired factorization, analogous to a Cholesky decomposition: $\hat{G} = (I - \hat{B})^{-1}\hat{\Lambda}^2(I - \hat{B})^{-T}$, where $\hat{B}$ is constrained to have zero diagonal and $\hat{\Lambda}^2$ is diagonal with positive entries. This is well known to exactly parametrize the set of covariance matrices; other parameterizations could of course be used.

The latent dimension can be determined via calculating the rank of observations, i.e., the number of non-zero singular values of sample covariance $\widehat{\text{Cov}}(X^{\tau})$; or in practice, the number of clearly non-zero singular values according to a given threshold. We optimize the objective with Adam (Kingma & Ba, 2015). This loss function is non-convex, as shown in an example in App. D.1.

**Step 2: Latent recovery using the identification mapping.** Given the estimates above, we recover latent variables via the following formula,

$$\hat{Z}^{\tau} = (\hat{D}^{\tau})^{-1}(\hat{A}^T\hat{A})^{-1}\hat{A}^T X^{\tau}, \qquad \tau = 1,\ldots,T. \quad (4)$$

This corresponds to projecting $X^{\tau}$ onto the estimated latent space followed by domain-wise rescaling. We analyse the

---

[1] We recall that a monomial matrix is a square matrix where each row and each column contains exactly one nonzero entry.

stability of source extraction, specifically the inversion of $\hat{A}^T \hat{A}$, in App. D.2. We analyze the convergence of our method and show that minimizing the covariance-matching objective leads to consistent recovery of the latent variables (proven in App. B.2.3):

**Theorem 4.1.** *We assume the same conditions as Thm. 3.2. Let $(\hat{A}, \{\hat{D}^\tau\}_{\tau=1}^{T}, \hat{G})$ be any global minimizer of the covariance-matching objective Eq. 3. We assume that for each domain $\tau \in [T]$, sample covariance $\widehat{\text{Cov}}(X^\tau)$ in Eq. 3 consistently estimates $\text{Cov}(\mathbf{X}^\tau)$. Then, the latent variables $\mathbf{Z}$ is consistently estimated by $(\hat{D}^\tau)^{-1}(\hat{A}^T \hat{A})^{-1}\hat{A}^T \mathbf{X}^\tau$ up to the following indeterminacies:*

- *Permutation, scaling, and domain-specific sign (Def. 2.1.(i)) in the general case;*

- *Permutation and scaling only (Def. 2.1.(ii)) if all entries of $D^\tau$ share the same sign across domains and $\hat{D}^\tau$ is constrained accordingly.*

**Connection with joint diagonalization.** Joint diagonalization (Belouchrani et al., 2002; Pham & Cardoso, 2001; Ablin et al., 2019) plays a major role in signal processing, blind source separation, and neuroscience, and has been widely used in the context of time series and spatial data. It can be viewed as a special form of MuDo-CoM in which the covariance matrices $\widehat{\text{Cov}}(X^\tau)$ and $\hat{G}$ in (3) are constrained to be diagonal, based on the assumption that the underlying $G$ is diagonal. In the classical joint diagonalization method, multiple covariance matrices corresponding to different domains are simultaneously factorized to recover a common mixing matrix, with identifiability arising from variability across domains. MuDo-CoM generalizes this idea by allowing the latent covariance to have *arbitrary dependency* and by explicitly *modeling domain-specific diagonal scalings*, while still utilizing cross-domain variability through covariance matching. In this sense, MuDo-CoM extends joint diagonalization from independent latent sources to dependent latent variables.

## 5. Related Work

Table 1 summarizes the most closely related approaches to our work. They include classical ICA (Hyvarinen, 1999), blind source separation via joint diagonization (Ablin et al., 2019), three-way tensor decompositions (Harshman et al., 1970), dependent factor analysis (Monti & Hyvärinen, 2018), and more recent lines of work that study identifiability of deep generative models and VAEs under various auxiliary conditions (Khemakhem et al., 2020). We summarize their key assumptions and limitations regarding latent dependence, the parametric structure of $\mathbf{Z}$, and the properties of the mixing matrix $A$, and discuss them, as well as other related methods, in detail in the following.

**(Non)linear ICA methods.** Linear ICA aims to recover latent sources from linear entangled measurements under the assumption that sources are statistically independent. Foundational identifiability results were introduced in Comon (1994); Cardoso (1998); Hyvärinen et al. (2001), and practical algorithms such as FastICA (Hyvarinen, 1999) remain widely used. Methods for multi-domain linear ICA are widely used in fMRI analysis, but they still assume independent, non-Gaussian components (Erhardt et al., 2011). Hyvärinen & Hoyer (2000); Bach & Jordan (2003a); Köster & Hyvärinen (2010) proposed models for dependencies of components, but no such method incorporates arbitrary linear correlations. For the non-linear ICA setting, Khemakhem et al. (2020); Kivva et al. (2022); Lachapelle et al. (2022) use auxiliary variables that parameterize latent distributions to recover sources. Temporal nonlinear ICA approaches use nonstationarity or temporal order (Hyvärinen et al., 2023). Some of these methods could be directly used for multi-domain data. However, all of the above approaches rely on (conditional) independence assumptions on the latent variables, whereas our setting allows arbitrary dependencies among the latent components.

**Blind source separation (BSS).** Current approaches to BSS rely on distinct structural or distributional assumptions about the latent variables, whereas our framework avoids imposing parametric constraints on their distributions and instead requires a linear mixing function together with domain-dependent scaling. For instance, non-negative matrix factorization and simplex-structured factorizations assume non-negativity of the observations, while bounded component analysis (Erdogan, 2012) restricts the latent support to a bounded geometric set. Similarly, tree-dependent component analysis (Bach & Jordan, 2003b) enforces a tree-structured dependency among latent variables, and dependent source separation method (Caiafa, 2012) assume linear dependencies between components. In contrast, our approach allows arbitrary latent distributions and dependency structures, trading these assumptions for structural constraints on the mixing process across domains. Another line of work in BSS, which is closer to our idea, is the use of nonstationarity. Pham & Cardoso (2001) proposed joint diagonalization of covariance matrices to separate Gaussian sources with nonstationary power (variance). While this was originally formulated in terms of time segments and non-stationarity, it is equivalent to a multi-domain method. Ablin et al. (2019) proposed a new quasi-Newton method for solving the joint diagonalization problem that is central in the computational implementation. These approaches target models where the latent covariance matrices are all diagonal, and thus, the latent variables are uncorrelated, which is in contrast to our setting.

*Table 1.* Comparison of closely related approaches in terms of assumptions on latent dependence, parametric assumptions on the latent variables $\mathbf{Z}$, and structural assumptions on the mixing matrix $A$. Both QNDIAG and PARAFAC implicitly assume that latent components are uncorrelated within each domain, because QNDIAG performs joint diagonalization of a collection of covariance matrices, and PARAFAC does not model cross-interactions between latent components. For more details, see App. E.2.

| Method | Dependence | Para. Assump. on Z | Struc. Assump. on $A$ |
|---|---|---|---|
| FastICA (Hyvarinen, 1999) | $\mathbf{Z}_i \perp \mathbf{Z}_j$ | non-gaussian | ●None |
| QNDIAG (Ablin et al., 2019) | $\text{cov}(\mathbf{Z}_i, \mathbf{Z}_j) = 0$ | ●None | ●None |
| PARAFAC-COV (Harshman et al., 1970) | $\text{cov}(\mathbf{Z}_i, \mathbf{Z}_j) = 0$ | ●None | ●None |
| iVAE (Khemakhem et al., 2020) | $\mathbf{Z}_i \perp \mathbf{Z}_j \vert U$ | Cond. exponential family | ●None |
| MHA (Monti & Hyvärinen, 2018) | ●None | ●None | Orthonormal + nonnegative |
| MuDo-CoM (Our method) | ●None | ●None | ●None |

**Tensor decompositions.** Some tensor decomposition approaches can be adapted to recover latent variables from linearly mixed observations collected across multiple domains, although precise identifiability analysis seems to be lacking. Multiway tensor decompositions such as PARAFAC/CANDECOMP (Harshman et al., 1970; Carroll & Chang, 1970) provide strong uniqueness guarantees under rank conditions, with modern implementations available in TensorLy. PARAFAC2 (Kiers et al., 1999) extends it to handle tensors with one mode varying in size across slices (e.g., differing numbers of rows), whereas PARAFAC assumes consistent dimensions across all modes. In our setup, each tensor corresponds to a covariance matrix of the same size, so PARAFAC2 would not add useful flexibility. Domanov & De Lathauwer (2013a;b); Sidiropoulos et al. (2017) have broadened the uniqueness theory. Among those, Harshman et al. (1970) is the closest to our setting: it can be interpreted as saying that the mixing matrices are allowed to vary across groups but are constrained to be identical up to column-wise scalings. When the group-specific scaling matrices are sufficiently distinct, the latent sources become identifiable by decomposing the set of covariance matrices. However, again, they require the sources to be uncorrelated.

**Causal representation learning** (CRL; Schölkopf et al., 2021) studies identifiability of latent variables from high-dimensional observations, when these variables are dependent, in particular due to causal relations between them. Current CRL methods require additional information or parametric assumptions, such as interventional data (Brehmer et al., 2022; Lippe et al., 2022; von Kügelgen et al., 2023; Buchholz et al., 2023; Zhang et al., 2023; Ahuja et al., 2023), temporal structure (Lippe et al., 2023a; Lachapelle et al., 2022; 2024; Li et al., 2025), multi-view settings (Yao et al., 2024; Ahuja et al., 2024; Morioka & Hyvarinen, 2024), hierarchical structures (Kong et al., 2023; 2024) or conditions on causal graphs (Zhang et al., 2024; Dong et al., 2024; Song et al., 2024). In particular, similarly to our method, some CRL works explicitly study linear observation models, providing identifiability guarantees under interventions, auxiliary variables, or sparse latent support (Squires et al.,

2023; Xu et al., 2024; Jin & Syrgkanis, 2024; Varici et al., 2025). In contrast, in our work, we do not use any causal interpretation of the variables nor make any assumption on why or how they might be dependent. For example, this can be due to causal relations, but also latent confounding or selection bias. When our method is applied to CRL, which assumes that the variables have a causal interpretation and the dependences are due to causal relations, scaling can be seen as a "global" soft intervention on all nodes simultaneously. In this setting, similar work like (Varici et al., 2025) could not be applied since it requires sufficiently diverse intervention targets, while (Zhang et al., 2024) would allow only for identifiability up to the intimate set of the causal variables.

## 6. Experimental Results

We first assess the performance of the proposed model using simulated data and then present an application to a pre-processed resting-state fMRI dataset (Monti et al., 2020). Details are in App. D.

### 6.1. Simulated Datasets

**Data generation.** We generate simulated datasets from the proposed model. The dependencies among the latent variables are created by an underlying structural causal model (SCM) like in CRL. We consider $n = \{5, 10, 20\}$ latent variables $\mathbf{Z}$. For each experiment, we sample a directed acyclic graph $\mathcal{D}$ from an Erdös-Rényi (ER)-$k$ model with $k \in \{0, 1, 2, 3\}$, where each ER-$k$ graph contains $n \cdot k$ directed edges. In particular, ER-0 corresponds to an empty graph, which implies independent latent variables. Given $\mathcal{D}$, we consider two types of SCM: i) a linear SCM where edge weights are sampled uniformly from $[-2, -0.5] \cup [0.5, 2]$ and we use three type of noises: standard Gaussian, exponential with scale 1 and Gumbel; ii) a nonlinear SCM, where we simulate a nonlinear function with a linear layer, a leaky-ReLU activation and a fully connected layer with 100 hidden units, where edge weights scaled by input dimension and noises are both sampled from standard Gaussian. To

*Table 2.* Results for the numerical experiments: Average d-MCC $\pm$ Std over 5 random seeds. The bold font indicates which parameters are varying in each block of rows and the best results among all methods.

| $n$ | $d$ | $T$ | $N$ | $k$ | SCMs | MuDo-CoM | FastICA | QNDIAG | PARAFAC | iVAE | MHA |
|---|---|---|---|---|---|---|---|---|---|---|---|
| **5** | $n$ | $2n$ | 5000 | 2 | Gauss | **0.999**±**.000** | 0.786±.056 | 0.814±.049 | 0.688±.151 | 0.808±.062 | 0.783±.033 |
| **10** | $n$ | $2n$ | 5000 | 2 | Gauss | **0.999**±**.001** | 0.783±.013 | 0.823±.019 | 0.477±.200 | 0.735±.035 | 0.726±.039 |
| **20** | $n$ | $2n$ | 5000 | 2 | Gauss | **0.998**±**.001** | 0.791±.021 | 0.835±.018 | 0.319±.081 | 0.711±.075 | 0.618±.018 |
| 10 | **1$n$** | $2n$ | 5000 | 2 | Gauss | **0.999**±**.001** | 0.783±.013 | 0.823±.019 | 0.477±.200 | 0.735±.035 | 0.726±.039 |
| 10 | **2$n$** | $2n$ | 5000 | 2 | Gauss | **0.999**±**.001** | 0.783±.013 | 0.823±.019 | 0.509±.167 | 0.734±.017 | 0.687±.027 |
| 10 | **4$n$** | $2n$ | 5000 | 2 | Gauss | **0.997**±**.004** | 0.783±.013 | 0.823±.019 | 0.464±.204 | 0.792±.022 | 0.689±.031 |
| 10 | **10$n$** | $2n$ | 5000 | 2 | Gauss | **0.999**±**.000** | 0.783±.013 | 0.823±.019 | 0.453±.189 | 0.806±.025 | 0.675±.051 |
| 10 | $n$ | **1$n$** | 5000 | 2 | Gauss | **0.998**±**.000** | 0.782±.035 | 0.820±.028 | 0.678±.145 | 0.766±.041 | 0.687±.014 |
| 10 | $n$ | **2$n$** | 5000 | 2 | Gauss | **0.999**±**.001** | 0.783±.013 | 0.823±.019 | 0.477±.200 | 0.735±.035 | 0.726±.039 |
| 10 | $n$ | **3$n$** | 5000 | 2 | Gauss | **0.991**±**.016** | 0.818±.018 | 0.857±.021 | 0.465±.191 | 0.759±.035 | 0.672±.018 |
| 10 | $n$ | $2n$ | **200** | 2 | Gauss | **0.900**±**.087** | 0.784±.013 | 0.823±.018 | 0.512±.187 | 0.662±.034 | 0.728±.038 |
| 10 | $n$ | $2n$ | **1000** | 2 | Gauss | **0.988**±**.014** | 0.784±.014 | 0.823±.019 | 0.581±.147 | 0.678±.013 | 0.727±.040 |
| 10 | $n$ | $2n$ | **5000** | 2 | Gauss | **0.999**±**.001** | 0.783±.013 | 0.823±.019 | 0.477±.200 | 0.735±.035 | 0.726±.039 |
| 10 | $n$ | $2n$ | **10000** | 2 | Gauss | **1.000**±**.000** | 0.783±.013 | 0.823±.019 | 0.518±.198 | 0.744±.050 | 0.727±.038 |
| 10 | $n$ | $2n$ | 5000 | **0** | Gauss | 0.994±.004 | **1.000**±**.000** | **1.000**±**.000** | 0.954±.091 | 0.914±.035 | 0.507±.027 |
| 10 | $n$ | $2n$ | 5000 | **1** | Gauss | **0.999**±**.001** | 0.850±.017 | 0.883±.010 | 0.583±.135 | 0.782±.047 | 0.635±.025 |
| 10 | $n$ | $2n$ | 5000 | **2** | Gauss | **0.999**±**.001** | 0.783±.013 | 0.823±.019 | 0.477±.200 | 0.735±.035 | 0.726±.039 |
| 10 | $n$ | $2n$ | 5000 | **3** | Gauss | **0.999**±**.001** | 0.745±.016 | 0.791±.009 | 0.474±.091 | 0.728±.025 | 0.776±.017 |
| 10 | $n$ | $2n$ | 5000 | 2 | **Gauss** | **0.999**±**.001** | 0.783±.013 | 0.823±.019 | 0.477±.200 | 0.735±.035 | 0.726±.039 |
| 10 | $n$ | $2n$ | 5000 | 2 | **Exp** | **0.991**±**.016** | 0.782±.014 | 0.823±.019 | 0.503±.164 | 0.723±.025 | 0.726±.039 |
| 10 | $n$ | $2n$ | 5000 | 2 | **Gumbel** | **0.996**±**.007** | 0.786±.017 | 0.823±.020 | 0.488±.228 | 0.746±.039 | 0.726±.039 |
| 10 | $n$ | $2n$ | 5000 | 2 | **Nonlinear** | **0.999**±**.001** | 0.857±.049 | 0.960±.011 | 0.527±.143 | 0.881±.028 | 0.587±.021 |

simulate the observed variables $\mathbf{X}^\tau$, we define the number of domains $T = \{1, 2, 3\} \times n$. In each domain $\tau \in [T]$, we first scale the latent variables by a diagonal domain-specific matrix $D^\tau$ with entries sampled uniformly in $[0.2, 1]$, and then apply a full-column mixing matrix $A$ to the latent variables $\mathbf{Z}$. The mixing matrix $A \in \mathbb{R}^{d \times n}$ is generated randomly depending on $d \in \{1, 2, 4, 10\} \times n$. For each domain $\tau$, we generate observed variables $\mathbf{X}^\tau = AD^\tau\mathbf{Z}$ sample sizes $N \in \{200, 1000, 5000, 10000\}$. We average the results for each setup over 5 seeds.

**Baselines.** We summarize the baselines in Tab. 1. As described in App. E.2, QNDIAG, PARAFAC-COV, MHA and iVAE can be directly used in multi-domain settings, while for ICA, we concatenate multi-domain data as a single dataset with a mixture model, which can ensure the non-Gaussianity is satisfied for ICA.

**Evaluation Metric.** We use three different evaluation metrics. The main one is Domain-Wise MCC (d-MCC) that we propose as a multi-domain version of the Mean Correlation Coefficient (Khemakhem et al., 2020). We compute MCC separately for each domain and report their average, capturing recovery up to permutation and sign inside each domain. This is to be contrasted with the standard MCC, which does not take the domain-specific sign indeterminacy into account, so it is inadequate in a multi-domain setting where the signs can vary between domains (Thm. 3.3). We also report the Amari distance (Amari et al., 1995), a standard ICA metric quantifying deviations of the estimated

unmixing matrix from the ground truth up to permutation and scaling. Details are in App. E.1.

**Results.** Tab. 2 reports the identification accuracy for different values of $n$, $d$, $T$, $N$, $k$, and SCMs. Our method consistently achieves (near-)perfect recovery across all settings. It has the best performance in all cases except the one that correpond to single-domain ICA. In contrast, all baselines except MHA assume independent or uncorrelated latents, and show clear degradation as there are more dependencies between the latents when the degree $k$ increases. Varying the latent dimension $n$ and observation dimension $d$ shows that our method, FastICA, and QNDIAG scale well with model size, while PARAFAC, iVAE, and MHA deteriorate considerably. W.r.t. sample size $N$, our estimator achieves high accuracy at $N = 200$, and increasing the sample size $N$ leads to perfect recovery. Finally, we show robustness across Gaussian, Exponential, and Gumbel noise models, as well as nonlinear SCMs. We provide more results, including MCC and Amari distance, in App. E.3.

In App. F we discuss our results when our assumptions are violated. In App. F.1, we show empirically that our method has a robust performance also in a noisy mixing setting. On the other hand, we do need all other assumptions. We provide an evaluation showing the necessity of linearity for the mixing function in App. F.2, showing that the performance become worse when the nonlinearity of mixing function increase (controlled by numbers of Leaky-ReLU layers). Similarly, we test the robustness of our method when the

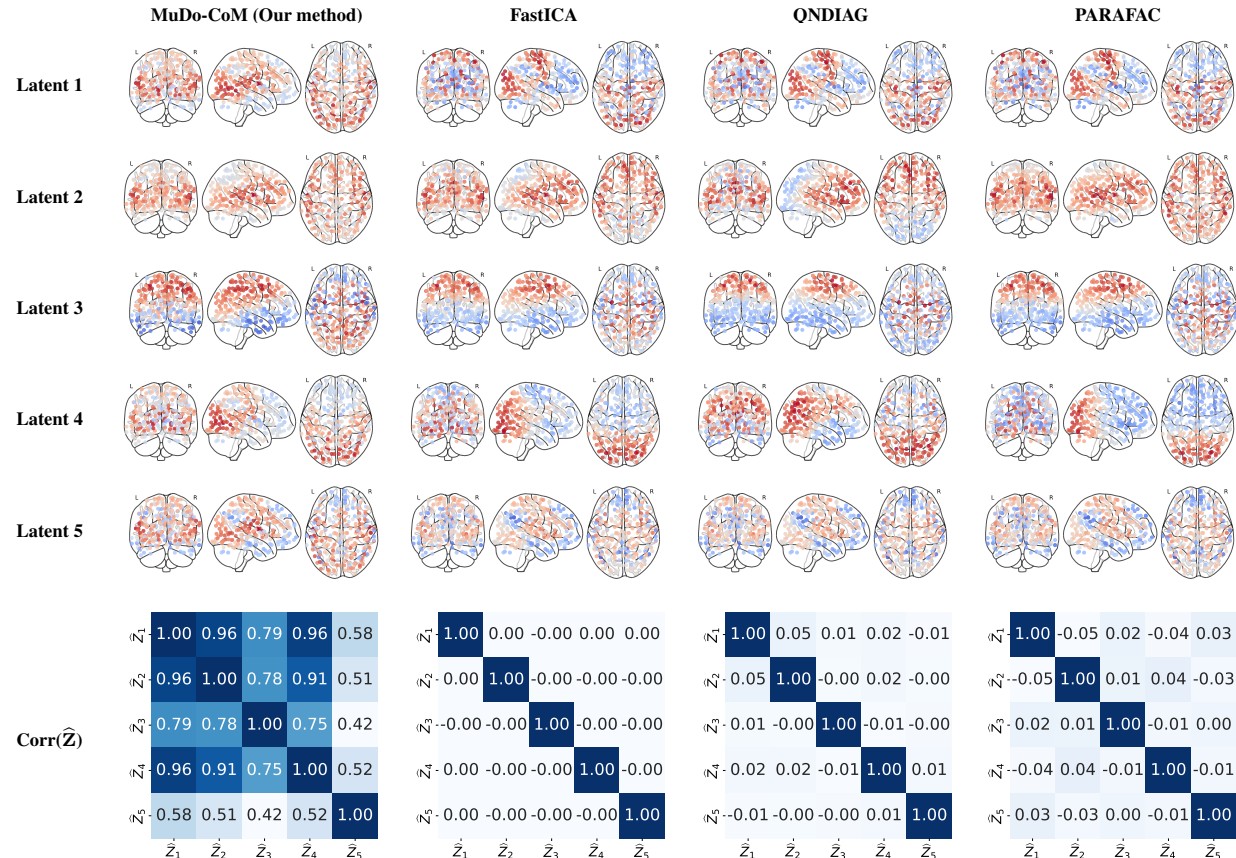

*Figure 2.* Visualization of the estimated components for the fMRI data for one random seed. Each column gives the five columns of the matrix $\hat{A}$ for one method, after aligning permutation and signs. Red means positive activation in a parcel, while blue means negative activity; the brain is viewed from three different angles. The last row shows the sample correlation matrix of the recovered variables $\hat{\mathbf{Z}}$.

assumption of invariant mixing $A$ and latent correlation matrix across domains in App. F.3 are violated, showing a reasonably robust performance to moderate deviations, with performance degrading gradually as the deviation increases. In App. F.4, we further provide two possible heuristics to test Ass. 3.1 (sufficient domain variability) empirically from observed data.

### 6.2. Real Dataset: CamCAN

**Dataset.** We use a publicly available preprocessed resting-state fMRI dataset by Monti et al. (2020), derived from the CamCAN cohort (Ogawa et al., 2018; Shafto et al., 2014; Taylor et al., 2017). A cortical parcellation based on resting-state functional connectivity analyses (Power et al., 2011) was used to define 264 distinct 10-mm diameter regions of interest (ROIs). The fMRI time courses were averaged across all voxels within each of the $d = 264$ ROI. We treat each of the $T = 565$ subjects as one domain. The dataset captures brain activity at $N = 261$ time points per subject. Following Monti & Hyvärinen (2018); Monti et al. (2020), we choose $n = 5$ as the latent dimension.

**Results.** In Figure 2, we show the learned components for our method and for the baseline methods. We plot the estimated columns of the mixing matrix $\hat{A}$, as each column presents the coefficients of one latent component, each corresponding to a spatial pattern of brain activity. For better visualization, we permute the components of the baseline methods to best align with the estimates of our method, potentially also switching their signs.

We further compute the sample correlation matrix of the estimated latent variables $\hat{\mathbf{Z}}$ as shown on the bottom row. This shows that our method finds components some of which are strongly correlated, even with a correlation coefficient of $0.96$. In contrast, the baseline methods find almost perfectly uncorrelated components, which is not surprising given that they assume the components to be uncorrelated.

Most learned latent components (LC) reflect activity in occipital, temporal and parietal brain areas, which is compatible with the prominent visual, auditory and somatomotor components of the well-known fMRI resting-state networks (Fox & Raichle, 2007; Seitzman et al., 2019). LC1 captures

a combination of activity in auditory and high-level visual cortical areas while LC2 seems more specific to temporal auditory areas with contributions from frontal cortical areas; yet, LC2 remains highly correlated with LC1. LC3 displays an interplay of superior and inferior halves of the entire cerebrum, which interestingly does not correspond to a known resting-state component or component set. LC4 again focuses on visual cortical areas with a weak contribution from somatomotor areas around the central sulcus and is highly correlated with LC1. LC5 reflects joint activity fluctuations of high-order visual, auditory and possibly also somatosensory cortices and displays a similar spatial pattern as LC1 although has a lower sample correlation with it. Tentatively, we might consider the combination of LC1, LC2, and LC4 as a bigger network of correlated components, which the baseline methods were unable to find.

To further investigate the stability of the learned components, we perform a bootstrap analysis with 20 resampled datasets, refit the model on the full dataset to obtain estimates of the latent components, and report the d-MCC$= 0.88 \pm 0.11$ between the bootstrap estimates and the reference estimate from the full dataset. This shows the stability of the estimation of the mixing matrices.

## 7. Conclusion and Limitation

In this work, we study a linear mixing model with dependent latent components across multiple data domains. We introduce identifiability results that generalize ICA-style identifiability to arbitrary dependence, based on the sufficient variability of domain-specific scalings. Building on these theoretical insights, we propose MuDo-CoM, a multi-domain covariance-matching estimation algorithm. Our approach generalizes classical non-stationarity-based second-order methods (Pham & Cardoso, 2001; Ablin et al., 2019) by allowing correlated latent variables. We validate the proposed method on both simulated and real-world fMRI datasets.

While our assumptions are relatively mild, several nontrivial gaps between theory and practice remain. We provide a detailed discussion in App. G. In particular, incorporating measurement noise, extending the model to nonlinear mixing functions, and developing more statistically efficient objectives are important directions for future work.

## Acknowledgements

The research of DX and SM was supported by the Air Force Office of Scientific Research under award number FA8655-22-1-7155. Any opinions, findings, and conclusions or recommendations expressed in this material are those of the author(s) and do not necessarily reflect the views of the United States Air Force. We also thank SURF for the support in using the Dutch National Supercomputer Snellius.

## Impact Statement

This paper presents work whose goal is to advance the field of Machine Learning, and specifically identifiable Representation Learning. There are many potential societal consequences of our work, none which we feel must be specifically highlighted here.

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

# A. Summary of Notation

Throughout this paper, we adopt the following conventions for mathematical notation: We use lowercase italic letters such as $x$ to denote scalars, bold lowercase letters such as $\mathbf{x}$ to denote deterministic vectors, bold uppercase letters such as $\mathbf{X}$ to denote random vectors, and uppercase italic letters such as $X$ to denote matrices, and calligraphic uppercase letters such as $\mathcal{X}$ to denote sets (e.g., a set of inputs or a domain). We list all specific notations in the following table:

| Symbol | Description |
|--------|-------------|
| $\tau$ | Domain index |
| $\mathbf{Z}$ | Latent variables |
| $\Lambda$ | Covariance matrix of the noise term |
| $\mathbf{X}^\tau$ | Observations from domain $\tau$ |
| $A$ | Linear mixing matrix |
| $D^\tau$ | Domain-specific scaling matrix for domain $\tau$ |
| $\Sigma^\tau$ | Covariance matrix of observations in domain $\tau$ |
| $G$ | Covariance matrix of latent components |
| $n$ | Dimension of the latent space |
| $d$ | Dimension of the observation space |
| $T$ | Number of domains |
| $N$ | Number of samples per domain |

# B. Proofs

## B.1. Justification of Assumptions

**Assumption 3.1.** [Sufficient variability of domain scalings] There exist $m = \binom{n+3}{4}$ domains $\tau_1, \ldots, \tau_m$, each with a domain-specific diagonal scaling matrix $D^\tau = \operatorname{diag}(d_1^\tau, \ldots, d_n^\tau)$, such that the matrix

$$
\begin{pmatrix}
\mathbf{d}_{(4)}^{\tau_1} & \mathbf{d}_{(3,1)}^{\tau_1} & \mathbf{d}_{(2,2)}^{\tau_1} & \mathbf{d}_{(2,1,1)}^{\tau_1} & \mathbf{d}_{(1,1,1,1)}^{\tau_1} \\
\mathbf{d}_{(4)}^{\tau_2} & \mathbf{d}_{(3,1)}^{\tau_2} & \mathbf{d}_{(2,2)}^{\tau_2} & \mathbf{d}_{(2,1,1)}^{\tau_2} & \mathbf{d}_{(1,1,1,1)}^{\tau_2} \\
\vdots & \vdots & \vdots & \vdots & \vdots \\
\mathbf{d}_{(4)}^{\tau_m} & \mathbf{d}_{(3,1)}^{\tau_m} & \mathbf{d}_{(2,2)}^{\tau_m} & \mathbf{d}_{(2,1,1)}^{\tau_m} & \mathbf{d}_{(1,1,1,1)}^{\tau_m}
\end{pmatrix}
\tag{2}
$$

is full-row rank, composed of the following sub-matrices:

$$
\begin{aligned}
\mathbf{d}_{(4)}^{\tau} &= \{(d_i^\tau)^4\}_{i=1}^n, \\
\mathbf{d}_{(3,1)}^{\tau} &= \{(d_i^\tau)^3 d_j^\tau\}_{i,j\in[n] \text{ s.t. } i\neq j}, \\
\mathbf{d}_{(2,2)}^{\tau} &= \{(d_i^\tau)^2(d_j^\tau)^2\}_{i,j\in[n], \text{ s.t. } i<j}, \\
\mathbf{d}_{(2,1,1)}^{\tau} &= \{(d_a^\tau)^2 d_b^\tau d_c^\tau\}_{a,b,c\in[n], \text{ s.t. all distinct, } b<c}, \\
\mathbf{d}_{(1,1,1,1)}^{\tau} &= \{d_a^\tau d_b^\tau d_c^\tau d_d^\tau\}_{a,b,c,d\in[n], \text{ s.t. } a<b<c<d}.
\end{aligned}
$$

In addition, $d_i^{\tau_j} \neq 0$ for all $i \in [n]$ and $j \in [m]$.

We first provide a simple derivation to explain why $m = \binom{n+3}{4}$. Then we provide a low-dimensional example below for $n = 2$ to show exactly what the "degree-4 monomials" matrix looks like.

**Why $m = \binom{n+3}{4}$?** The total number of columns equals the number of distinct monomials of total degree four, which can be counted by grouping monomials according to their exponent patterns:

$$
\begin{aligned}
\#(4) &: \ n, \\
\#(3,1) &: \ n(n-1), \\
\#(2,2) &: \ \binom{n}{2}, \\
\#(2,1,1) &: \ n\binom{n-1}{2}, \\
\#(1,1,1,1) &: \ \binom{n}{4}.
\end{aligned}
$$

Summing these contributions yields

$$
n + n(n-1) + \binom{n}{2} + n\binom{n-1}{2} + \binom{n}{4} = \binom{n+3}{4},
$$

which is exactly the number of all possible homogeneous monomials of degree 4 with $n$ variables.

**Example B.1** ($n = 2$). When $n = 2$, the set of monomials of total degree four in $(d_1^\tau, d_2^\tau)$ is

$$
(d_1^\tau)^4, \quad (d_1^\tau)^3 d_2^\tau, \quad (d_1^\tau)^2 (d_2^\tau)^2, \quad d_1^\tau (d_2^\tau)^3, \quad (d_2^\tau)^4.
$$

Hence $m = \binom{2+3}{4} = 5$ domains are sufficient, and the full-rank condition in Assumption 3.1 reduces to requiring that the $5 \times 5$ matrix

$$
\begin{pmatrix}
(d_1^{\tau_1})^4 & (d_1^{\tau_1})^3 d_2^{\tau_1} & (d_1^{\tau_1})^2 (d_2^{\tau_1})^2 & d_1^{\tau_1} (d_2^{\tau_1})^3 & (d_2^{\tau_1})^4 \\
(d_1^{\tau_2})^4 & (d_1^{\tau_2})^3 d_2^{\tau_2} & (d_1^{\tau_2})^2 (d_2^{\tau_2})^2 & d_1^{\tau_2} (d_2^{\tau_2})^3 & (d_2^{\tau_2})^4 \\
\vdots & \vdots & \vdots & \vdots & \vdots \\
(d_1^{\tau_5})^4 & (d_1^{\tau_5})^3 d_2^{\tau_5} & (d_1^{\tau_5})^2 (d_2^{\tau_5})^2 & d_1^{\tau_5} (d_2^{\tau_5})^3 & (d_2^{\tau_5})^4
\end{pmatrix}
$$

is invertible, with all entries $d_i^{\tau_j} \neq 0$.

## B.2. Identifiability Theorems

### B.2.1. PROOF OF THEOREM 3.2

**Theorem 3.2.** *Consider the setting described in Sec. 2. Suppose Ass. 3.1 is satisfied. Then, the domain-invariant mixing matrix A is identifiable up to permutation and scaling (including sign switches), i.e., A is recovered up to right multiplication by a monomial matrix* [2].

Before we prove Theorem 3.2, we first introduce several basic lemmas that are useful to drive the proof.

**Lemma B.2.** *Let $X \in \mathbb{R}^{n \times n}$ be full rank. If each row $X_{a\cdot}$ contains at most one nonzero entry, then $X$ is a monomial matrix.*

*Proof.* Since $X$ is full rank, no row of $X$ can be identically zero, otherwise $\mathrm{rank}(X) \leq n-1$. Together with the assumption that each row has at most one nonzero entry, this implies that each row has exactly one nonzero entry. Hence there exists a map $\sigma : [n] \to [n]$ and nonzero scalars $\lambda_a \neq 0$ such that

$$
X_{aj} = \begin{cases} \lambda_a, & j = \sigma(a), \\ 0, & \text{otherwise.} \end{cases}
$$

Now we show that $\sigma$ must be injective. We prove this by contradiction. Suppose not, then there exist $a \neq b$ with $\sigma(a) = \sigma(b) = j$. In that case, columns $j$ is the only column where rows $a$ and $b$ may have nonzero entries, so the two rows are linearly dependent since

$$
\lambda_b X_{a\cdot} - \lambda_a X_{b\cdot} = 0,
$$

---

[2]We recall that a monomial matrix is a square matrix where each row and each column contains exactly one nonzero entry.

contradicting $\text{rank}(X) = n$. Therefore $\sigma$ is an injective function. And since $\sigma()$ maps domain $[n]$ to itself, $\sigma$ is a permutation of $[n]$.

Let $P$ be the permutation matrix with $P_{a,\sigma(a)} = 1$, and let $\Lambda = \text{diag}(\lambda_1, \ldots, \lambda_n)$ with $\lambda_a \neq 0$. Then

$$X = \Lambda P,$$

which is a monomial matrix. $\qquad \square$

**Lemma B.3.** *Let $\Sigma \in \mathbb{R}^{n \times n}$ be a positive definite matrix. Then all diagonal entries of $\Sigma$ are strictly positive.*

*Proof.* Since $\Sigma$ is positive definite, by definition, we have

$$x^T \Sigma x > 0 \quad \text{for all } x \in \mathbb{R}^n \setminus \{\mathbf{0}\}. \tag{5}$$

Let $e_i$ be the $i$-th standard basis vector. Because $e_i \neq 0$, it follows that

$$e_i^T \Sigma e_i > 0. \tag{6}$$

We know

$$e_i^T \Sigma e_i = \Sigma_{ii}. \tag{7}$$

Therefore, $\Sigma_{ii} > 0$ for all $i \in [n]$, which means every diagonal entry of $\Sigma$ is non-zero. $\qquad \square$

The following lemma is the essential element for the proof of Theorem 3.2. The intuition behind it is that if a matrix $X$ can be sandwiched between many independent diagonal scalings and still induce the same covariance transformation, i.e., the transformation from covariance $G$ to covariance $\hat{G}$, then $X$ cannot mix coordinates at all. In other words, it can only permute and rescale them.

**Lemma B.4.** *Let $G, \hat{G} \in \mathbb{R}^{n \times n}$ be covariance matrices with nonzero diagonal entries, and let $X \in \mathbb{R}^{n \times n}$ be full-rank. Suppose we have $m$ diagonal matrices $D^i \in \mathbb{R}^{n \times n}$ satisfy Assumption 3.1.*

*If there exists a family of diagonal matrices $\hat{D}^i$, $i \in [m]$, such that*

$$\hat{G} = \left(D^i X (\hat{D}^i)^{-1}\right)^T G \left(D^i X (\hat{D}^i)^{-1}\right), \qquad \forall i \in [m]. \tag{8}$$

*Then $X$ is a monomial matrix.*

*Proof.* Let

$$D^i = \text{diag}(d_1^i, \ldots, d_n^i), \qquad \hat{D}^i = \text{diag}(\hat{d}_1^i, \ldots, \hat{d}_n^i). \tag{9}$$

We use $x_{pq}$ to denote the $(p, q)$-entry of $X$. Expand Equation (8) entry-wise out and we get:

$$\hat{G}_{pq} = (\hat{d}_p^i \hat{d}_q^i)^{-1} \sum_{r=1}^n \sum_{s=1}^n d_r^i d_s^i G_{rs} x_{rp} x_{sq}. \tag{10}$$

Since by condition $\hat{G}$ has nonzero diagonal entries, in particular $\hat{G}_{pp} \neq 0$ and $\hat{G}_{qq} \neq 0$. We consider the ratio $\frac{(\hat{G}_{pq})^2}{\hat{G}_{pp}\hat{G}_{qq}}$. Substituting (10) into this expression gives

$$\frac{(\hat{G}_{pq})^2}{\hat{G}_{pp}\hat{G}_{qq}} = \frac{(\hat{d}_p^i \hat{d}_q^i)^2}{(\hat{d}_p^i)^2 (\hat{d}_q^i)^2} \frac{\left(\sum_{r,s} d_r^i d_s^i G_{rs} x_{rp} x_{sq}\right)^2}{\left(\sum_{r,s} d_r^i d_s^i G_{rs} x_{rp} x_{sp}\right)\left(\sum_{r,s} d_r^i d_s^i G_{rs} x_{rq} x_{sq}\right)}. \tag{11}$$

Note that the first term on the right-hand side, i.e., the prefactor, is 1. Define $C_{pq} := \frac{(\hat{G}_{pq})^2}{\hat{G}_{pp}\hat{G}_{qq}}$, which is independent of $i$. Then the equation becomes

$$C_{pq} = \frac{\left(\sum_{r,s} d_r^i d_s^i G_{rs} x_{rp} x_{sq}\right)^2}{\left(\sum_{r,s} d_r^i d_s^i G_{rs} x_{rp} x_{sp}\right)\left(\sum_{r,s} d_r^i d_s^i G_{rs} x_{rq} x_{sq}\right)} \tag{12}$$

Expanding both numerator and denominator yields

$$C_{pq} = \frac{\sum_{r=1}^n \sum_{s=1}^n \sum_{k=1}^n \sum_{t=1}^n d_r^i d_s^i d_k^i d_t^i G_{rs} G_{kt} x_{rp} x_{sq} x_{kp} x_{tq}}{\sum_{r=1}^n \sum_{s=1}^n \sum_{k=1}^n \sum_{t=1}^n d_r^i d_s^i d_k^i d_t^i G_{rs} G_{kt} x_{rp} x_{sp} x_{kq} x_{tq}}. \tag{13}$$

Multiplying both sides by the denominator and rearranging terms, we obtain

$$\sum_{r=1}^n \sum_{s=1}^n \sum_{k=1}^n \sum_{t=1}^n d_r^i d_s^i d_k^i d_t^i G_{rs} G_{kt} x_{rp} x_{tq} (x_{sq} x_{kp} - C_{pq} x_{sp} x_{kq}) = 0. \tag{14}$$

We rewrite it in a form as a linear combination of degree-4 monomials in the scaling vector $(d_1^i, \ldots, d_n^i)$. In other words, combine the terms having the same $d_r^i d_s^i d_k^i d_t^i$. Then we get

$$0 = \sum_{a=1}^n \beta_a^{(4)} (d_a^i)^4 + \sum_{\substack{a,b=1 \\ a \neq b}}^n \beta_{ab}^{(3,1)} (d_a^i)^3 d_b^i + \sum_{1 \leq a < b \leq n} \beta_{ab}^{(2,2)} (d_a^i)^2 (d_b^i)^2$$

$$+ \sum_{\substack{a,b,c=1 \\ \text{all distinct}}}^n \beta_{abc}^{(2,1,1)} (d_a^i)^2 d_b^i d_c^i + \sum_{1 \leq a < b < c < d \leq n} \beta_{abcd}^{(1,1,1,1)} d_a^i d_b^i d_c^i d_d^i, \tag{15}$$

where the $\beta^{(\cdot)}$ are the coefficients that depend on the $G$, $X$, and the constants $C_{pq}$. Each coefficient is obtained by collecting all terms in (14) that correspond to the same degree-4 monomial in $(d_1^i, \ldots, d_n^i)$.

We rewrite Equation (15) in matrix form by collecting all degree-4 monomials of $(d_1^i, \ldots, d_n^i)$ into a feature vector

$$\phi(d^i) = \left[\mathbf{d}_{(4)}^i, \mathbf{d}_{(3,1)}^i, \mathbf{d}_{(2,2)}^i, \mathbf{d}_{(2,1,1)}^i, \mathbf{d}_{(1,1,1,1)}^i\right]^T,$$

and stacking the corresponding coefficients into a vector $\boldsymbol{\beta} = \left[\beta^{(4)}, \beta^{(3,1)}, \beta^{(2,2)}, \beta^{(2,1,1)}, \beta^{(1,1,1,1)}\right]^T$. This leads to the following form of vector production:

$$\phi(d^i)^T \boldsymbol{\beta} = 0, \qquad \forall i \in [m]. \tag{16}$$

Stacking these equations across all $m$ domains gives the linear system

$$\Phi \boldsymbol{\beta} = \mathbf{0},$$

where $\Phi \in \mathbb{R}^{m \times \binom{n+3}{4}}$ is the matrix whose $i$-th row is $\phi(d^i)^T$. By Assumption 3.1, the matrix $\Phi$ is full rank. It follows that the only solution is $\boldsymbol{\beta} = \mathbf{0}$.

Consider the first $n$ terms in $\boldsymbol{\beta}$, i.e. coefficient $\beta_a^{(4)}$, $a = 1, ..., n$, it is the case when $r = s = k = t = a$ in Equation (14). Thus we have

$$G_{aa}^2 x_{ap}^2 x_{aq}^2 (1 - C_{pq}) = 0, \forall a \in [n]. \tag{17}$$

Since we know $C_{pq} := \frac{(\hat{G}_{pq})^2}{\hat{G}_{pp}\hat{G}_{qq}}$ and $\hat{G}$ is a covariance matrix, by Cauchy–Schwarz inequality, we have $C_{pq} < 1$ when $p \neq q$. Thus, we can derive

$$G_{aa}^2 x_{ap}^2 x_{aq}^2 = 0, \qquad \forall a \in [n], \qquad \forall p \neq q. \tag{18}$$

As $G_{aa} \neq 0$, we can further simplify the equation and get

$$x_{ap}x_{aq} = 0\,, \qquad \forall a \in [n]\,, \qquad \forall p \neq q. \tag{19}$$

We first fix $a \in [n]$, Equation (19) implies that each row $X_{a.}$ has at most one non-zero element. Since $X$ is full-rank, by Lemma B.2, we can conclude that $X$ is monomial.

$\square$

Now we are ready to prove Theorem 3.2.

*Proof.* Suppose there is another set of latent variables $\hat{\mathbf{Z}}$ can generate the same $\mathbf{X}^{\tau}$ for all $\tau \in [T]$, following the same data-generating process as described in Section 2 with a new set of paramaters $\{\hat{A}, \hat{D}^{\tau}, \hat{G}\}$, where $\hat{D}^{\tau}$ are domain-specific scaling matrix with nonzero entries, $\hat{G}$ are definite positive covariance matrix of $\hat{\mathbf{Z}}$, $\hat{A}$ are full-column-rank mixing matrix, then we have,

$$AD^{\tau}\mathbf{Z} = \mathbf{X}^{\tau} = \hat{A}\hat{D}^{\tau}\hat{\mathbf{Z}}, \forall \tau \in [T], \tag{20}$$

where we slightly abuse notation by using $=$ to denote equality in distribution.

To prove $A$ is recovered up to right multiplication by a signed monomial matrix, i.e., there exists a monomial matrix $M \in \mathbb{R}^{n \times n}$, such that

$$\hat{A} = AM \tag{21}$$
$$M^{-1} = (\hat{A}^T\hat{A})^{-1}\hat{A}^T A. \tag{22}$$

Since $M^{-1}$ is a monomial matrix as well, it is equivalent to show $(\hat{A}^T\hat{A})^{-1}\hat{A}^T A$ is a monomial matrix. We aim to prove this in the next step.

Based on Equation (20), take covariance on both sides and then we have

$$Cov(\mathbf{X}^{\tau}) = AD^{\tau}GD^{\tau}A^T = \hat{A}\hat{D}^{\tau}\hat{G}\hat{D}^{\tau}\hat{A}^T. \tag{23}$$

Since all entries of the diagonal matrix $\hat{D}^{\tau}$ are nonzero, all columns are linearly independent of each other. This implies $\hat{D}^{\tau}$ is invertible. Rearrange Equation (23) then we have

$$\hat{G} = (\hat{D}^{\tau})^{-1}(\hat{A}^T\hat{A})^{-1}\hat{A}^T AD^{\tau}GD^{\tau}A^T\hat{A}(\hat{A}^T\hat{A})^{-1}(\hat{D}^{\tau})^{-1}, \tag{24}$$
$$\hat{G} = \left(D^{\tau}A^T\hat{A}(\hat{A}^T\hat{A})^{-1}(\hat{D}^{\tau})^{-1}\right)^T G \left(D^{\tau}A^T\hat{A}(\hat{A}^T\hat{A})^{-1}(\hat{D}^{\tau})^{-1}\right). \tag{25}$$

Since both $G$ and $\hat{G}$ are definite positive matrices, by Lemma B.3, we know all diagonal entries of them are nonzero. Besides, we know $A^T\hat{A}(\hat{A}^T\hat{A})^{-1}$ must be full-rank matrix since all $\hat{G}$, $G$, $D^{\tau}$ and $\hat{D}^{\tau}$ are full-rank in Equation (25). All conditions in Lemma B.4 got satisfied, and we can apply it to Equation (25). Therefore, we conclude that $A^T\hat{A}(\hat{A}^T\hat{A})^{-1}$ is a monomial matrix. The proof is completed.

$\square$

### B.2.2. PROOF OF THEOREM 3.3

**Theorem 3.3.** *We assume the same conditions as Thm. 3.2. Then, the domain-specific mixing $\{AD^{\tau}\}_{\tau=1}^T$ can be identified up to right multiplication by a (domain-independent) monomial matrix and a domain-related sign matrix $S^{\tau}$ for all $\tau \in [T]$, leading to the identifiability of $\mathbf{Z}$ up to permutation, scaling, and domain-specific sign (Def. 2.1.(i)).*

*Moreover, if all entries of $D^{\tau}$ share the same sign for every $\tau \in [T]$, then $\mathbf{Z}$ is identifiable up to permutation and scaling for all domains $\tau \in [T]$ (Def. 2.1.(ii)).*

*Proof.* To prove $AD^\tau$ is recovered up to right multiplication by a monomial matrix with a domain-specific sign matrix, i.e., there exists a monomial matrix $M \in \mathbb{R}^{n \times n}$ and sign matrix $S^\tau$ with diagonal entries from $\{-1, 1\}$, such that

$$\hat{A}\hat{D}^\tau = AD^\tau S^\tau M \tag{26}$$

$$M^{-1}(S^\tau)^{-1} = (\hat{D}^\tau)^{-1}(\hat{A}^T\hat{A})^{-1}\hat{A}^T AD^\tau. \tag{27}$$

Since $M^{-1}$ is a monomial matrix and $(S^\tau)^{-1}$ is a sign matrix still, it is equivalent to show $(\hat{D}^\tau)^{-1}(\hat{A}^T\hat{A})^{-1}\hat{A}^T AD^\tau$ is a monomial matrix times a domain-specific sign matrix. We aim to prove this in the next step.

From Theorem 3.2, we know that $(\hat{A}^T\hat{A})^{-1}\hat{A}^T A$ is a monomial matrix, which means there exists a permutation matrix $P$ and diagonal matrix $E$ such that

$$(\hat{A}^T\hat{A})^{-1}\hat{A}^T A = PE \tag{28}$$

$$\Rightarrow \quad (\hat{D}^\tau)^{-1}(\hat{A}^T\hat{A})^{-1}\hat{A}^T AD^\tau = (\hat{D}^\tau)^{-1}PED^\tau = PP^T(\hat{D}^\tau)^{-1}PED^\tau. \tag{29}$$

Define $E^\tau := P^T(\hat{D}^\tau)^{-1}PED^\tau$, which is a diagonal matrix related to domain index $\tau$. Then we have

$$(\hat{D}^\tau)^{-1}(\hat{A}^T\hat{A})^{-1}\hat{A}^T AD^\tau = PE^\tau. \tag{30}$$

Repeat the same derivation as in Theorem 3.2 and we can get Equation (25) as well. Substitute Equation (30) into it then we have

$$\hat{G} = PE^\tau G(PE^\tau)^T \tag{31}$$

Denote the permutation induced by the permutation matrix $P$ as $\pi : [n] \to [n]$. Considering the diagonal entries on both sides of Equation (31), we get

$$\hat{G}_{\pi(i)\pi(i)} = (E_i^\tau)^2 G_{ii} \tag{32}$$

$$\Rightarrow E_i^\tau = \pm\sqrt{\hat{G}_{\pi(i)\pi(i)}G_{ii}^{-1}}. \tag{33}$$

Define a diagonal matrix $L = diag\left(\sqrt{\hat{G}_{\pi(1)\pi(1)}G_{11}^{-1}}, \sqrt{\hat{G}_{\pi(2)\pi(2)}G_{22}^{-1}}, ..., \sqrt{\hat{G}_{\pi(n)\pi(n)}G_{nn}^{-1}}\right)$. Then, by Equation (33), we have

$$E^\tau = LS^\tau, \qquad \forall \tau \in [T], \tag{34}$$

where $S^\tau$ is a domain-specific sign matrix. Substitute Equation (34) into Equation (30), we finally get

$$(\hat{D}^\tau)^{-1}(\hat{A}^T\hat{A})^{-1}\hat{A}^T AD^\tau = PLS^\tau, \qquad \forall \tau \in [T]. \tag{35}$$

The proof of part (i) is completed.

For part (ii), we additionally assume all $D^\tau$ are sharing the same sign, as well as for $\hat{D}^\tau$, which means for all $i \in [n]$, $E_i^\tau$ must be the same sign across $\tau \in [T]$ in Equation (30). This implies that only one value in Equation (33) can be taken for each $i \in [n]$. Thus, we can find a sign matrix $S$, such that

$$S := S^\tau, \forall \tau \in [T]. \tag{36}$$

Consequently,

$$(\hat{D}^\tau)^{-1}(\hat{A}^T\hat{A})^{-1}\hat{A}^T AD^\tau = PLS, \qquad \forall \tau \in [T]. \tag{37}$$

The proof of part (ii) is completed. $\qquad\square$

B.2.3. PROOF OF THEOREM 4.1

**Theorem 4.1.** *We assume the same conditions as Thm. 3.2. Let $(\hat{A}, \{\hat{D}^\tau\}_{\tau=1}^T, \hat{G})$ be any global minimizer of the covariance-matching objective Eq. 3. We assume that for each domain $\tau \in [T]$, sample covariance $\widehat{\text{Cov}}(X^\tau)$ in Eq. 3 consistently estimates $\text{Cov}(\mathbf{X}^\tau)$. Then, the latent variables $\mathbf{Z}$ is consistently estimated by $(\hat{D}^\tau)^{-1}(\hat{A}^T\hat{A})^{-1}\hat{A}^T\mathbf{X}^\tau$ up to the following indeterminacies:*

- *Permutation, scaling, and domain-specific sign (Def. 2.1.(i)) in the general case;*

- *Permutation and scaling only (Def. 2.1.(ii)) if all entries of $D^\tau$ share the same sign across domains and $\hat{D}^\tau$ is constrained accordingly.*

*Proof.* Denote the loss function in Equation (3) as $\widehat{\mathcal{L}}_N$, where $N$ is the sample number per domain (rows number of $X^\tau$). Let $(\hat{A}_N, \{\hat{D}_N^\tau\}_{\tau=1}^T, \hat{G}_N)$ be the global minimizer of $\widehat{\mathcal{L}}_N$, and define the fitted matrices as

$$\hat{\Sigma}_N^\tau := \hat{A}_N \hat{D}_N^\tau \hat{G}_N \hat{D}_N^\tau \hat{A}_N^T. \tag{38}$$

Let $\Sigma^\tau := \text{Cov}(\mathbf{X}^\tau)$, i.e., the ground truth covariance matrix of observation from domain $\tau$. Since $\mathbf{X}^\tau$ is generated via the process discribed in Section 2, we have

$$\Sigma^\tau = AD^\tau GD^\tau A^T. \tag{39}$$

Since $(\hat{A}_N, \{\hat{D}_N^\tau\}_{\tau=1}^T, \hat{G}_N)$ minimizes $\widehat{\mathcal{L}}_N$,

$$0 \le \widehat{\mathcal{L}}_N(\hat{A}_N, \{\hat{D}_N^\tau\}_{\tau=1}^T, \hat{G}_N) \le \widehat{\mathcal{L}}_N(A, \{D^\tau\}_{\tau=1}^T, G) = \sum_{\tau=1}^T \left\| \widehat{\text{Cov}}(X^\tau) - \Sigma^\tau \right\|_F^2. \tag{40}$$

Since $\widehat{\text{Cov}}(X^\tau) \xrightarrow{P} \Sigma^\tau$ for each $\tau$ and $T$ is fixed, the right-hand side converges to zero in probability. Hence,

$$\widehat{\mathcal{L}}(\hat{A}_N, \{\hat{D}_N^\tau\}, \hat{G}_N) \xrightarrow{P} 0. \tag{41}$$

By definition of the loss function, we have

$$\widehat{\mathcal{L}}(\hat{A}_N, \{\hat{D}_N^\tau\}, \hat{G}_N) = \sum_{\tau=1}^T \left\| \widehat{\text{Cov}}(X^\tau) - \hat{\Sigma}_N^\tau \right\|_F^2. \tag{42}$$

Connecting it with Equation (41), we get

$$\sum_{\tau=1}^T \left\| \widehat{\text{Cov}}(X^\tau) - \hat{\Sigma}_N^\tau \right\|_F^2 \xrightarrow{P} 0. \tag{43}$$

Due to the nonnegativity of each term, we can further derive

$$\left\| \widehat{\text{Cov}}(X^\tau) - \hat{\Sigma}_N^\tau \right\|_F^2 \xrightarrow{P} 0, \qquad \forall \tau \in [T]. \tag{44}$$

Finally, we aim to show that $\hat{\Sigma}_N^\tau$ is a consistent estimator of $\Sigma^\tau$. By the triangular inequality,

$$0 \le \|\hat{\Sigma}_N^\tau - \Sigma^\tau\|_F \le \|\hat{\Sigma}_N^\tau - \widehat{\text{Cov}}(X^\tau)\|_F + \|\widehat{\text{Cov}}(X^\tau) - \Sigma^\tau\|_F. \tag{45}$$

The first term converges to 0 in probability by Equation (44) and the second term converges to 0 in probability by condition. Thus,

$$\hat{A}_N \hat{D}_N^\tau \hat{G}_N \hat{D}_N^\tau \hat{A}_N^T = \hat{\Sigma}_N^\tau \xrightarrow{P} \Sigma^\tau, \qquad \forall \tau \in [T]. \tag{46}$$

Assume we have infinite samples, i.e., $N \to \infty$, then after converge, $\hat{A}_N \hat{D}_N^\tau \hat{G}_N \hat{D}_N^\tau \hat{A}_N^T$ will equal to $\Sigma^\tau$. The following proof is basically the same as that of Theorem 3.3 (Appendix B.2.2), which leads to the conclusion that the recovered representations are equal to the true latents up to the identifiability class (permutation, scaling, and domain-specific sign). Under the additional sign-consistency constraint on $D^\tau$ (and the corresponding restriction on $\hat{D}^\tau$), domain-specific sign flips are ruled out and the ambiguity reduces to the classical permutation-and-scaling indeterminacy. $\square$

## C. Intrinsic Structure of Latent Components Z

In this section, we discuss potential prior knowledge about the intrinsic structure of the latent components $\mathbf{Z}$, which can help reduce the number of domains required for identifiability in the theorem.

**Uncorrelated latent components.** A particularly important special case is when the latent components are uncorrelated, i.e., the covariance matrix $G = \text{Cov}(\mathbf{Z})$ is diagonal. This setting corresponds to the classical assumptions of independent component analysis (ICA) and related blind source separation methods, e.g, Joint Diagonalization (Belouchrani et al., 2002; Pham & Cardoso, 2001). In this case, the covariance matrices across domains take the form

$$\text{Cov}(X^\tau) = AD^\tau GD^\tau A^\top,$$

where $G$ is diagonal and $D^\tau$ is domain-specific and diagonal.

Under this structure, the matrices $\{\text{Cov}(X^\tau)\}_{\tau=1}^T$ are jointly diagonalizable by the same linear transformation, up to permutation and scaling. As a result, identifiability of the mixing matrix $A$ can be achieved through classical joint diagonalization techniques, and the required number of domains is $n$, which is significantly smaller than in the fully dependent case.

From the perspective of our framework, joint diagonalization is a special case in which the latent covariance matrix has no off-diagonal entries. Incorporating this structural assumption into our analysis causes all mixed-coefficient terms $\beta^{(3,1)}, \beta^{(2,2)}, \beta^{(2,1,1)}$, and $\beta^{(1,1,1,1)}$ vanish to zero, so that Equation (15) reduces to a sum of $n$ terms only:

$$0 = \sum_{a=1}^n \beta_a^{(4)} (d_a^i)^4. \tag{47}$$

Therefore, only $n$ domains are required to ensure that Equation (47) has only one solution (equal to zero), which significantly reduces the domain numbers compared to the dependent case. This result is also consistent with the Joint Diagonization result.

## D. Implementation Details and Licences

This section provides further details about the experiment implementation in Section 6. The implementation is built on the open-source code by Zheng et al. (2018), released under the Apache 2.0 License, which we used to generate latent samples via the structural model. We list all the hyperparameters we used for training in Table 3, and provide all the code for our method, and the experiments at this [GitHub repository].

*Table 3.* Parameters for experiments results in Sec. 6 and App. E.

|                  | Numerical | CamCAN  |
| ---------------- | --------- | ------- |
| Optimizer        | Adam      | Adam    |
| Learning rate    | 5e-3      | 5e-3    |
| # Seeds          | 5         | 5 or 20 |
| # Iterations     | 50000     | 10000   |
| # Initialization | 3         | 20      |

### D.1. (Non)-Convexity of the Loss Function

Our loss function is not convex with respect to $\hat{A}$, $\hat{D}^\tau$, and $\hat{\Lambda}$ and $\hat{B}$, so it can in principle get stuck in local minima. We provide a simple example to show the nonconvexity:

**Example D.1.** Let $dim(X) = dim(Z) = 1$, domain number $T = 1$, observed covariance $Cov(X) := C > 0$. In this case $loss = (C - (AD\Lambda)^2)^2$. Note that here $C, A, D, \Lambda$ are all scalars. If we fix nonzero $D, \Lambda$, then we can regard the loss as a function of $A$, $Loss(A) = (D\Lambda)^4 A^4 - 2C(D\Lambda)^2 A^2 + C^2$. Taking the second derivative at $A = 0$, we get $-4C(D\Lambda)^2 < 0$. Thus the loss is not convex.

This matches our choice to use multiple random initializations to escape bad local minima (3 reinitializations used in Table 2), and keep the best run (lowest loss).

## D.2. Analysis of the Stability of Source Extraction

From a theoretical point of view, to ensure the identifiability of latent, $AA^T$ must be invertible. The reason is that in identifiability theory, a common and often necessary assumption is an injective mixing function, i.e., a full column-rank $A$, because if two distinct latent values $\mathbf{z}_1$ and $\mathbf{z}_2$ get mapped to the same observation $\mathbf{x}$, then generally it is impossible to distinguish them from $\mathbf{x}$ only. In our setup, this is equivalent to a full column-rank $A$, which implies a full column-rank $\hat{A}$ due to perfect covariance matching. Then, the stability of source extraction has already been sufficiently tested by computing MCC, which needs to use $(\hat{A}\hat{A}^T)^{-1}$ to compute $\hat{Z}$, in Tab2.

# E. Additional Experimental Results

## E.1. Metrics

**Evaluation Metrics.**   We use three metrics to assess the performance of our method in recovering the latent causal variables.

**i) Mean Correlation Coefficient (MCC).** Following previous work (Hyvarinen et al., 2019; Khemakhem et al., 2020), we report the *Mean Correlation Coefficient* (MCC) to evaluate how well the learned representations match the ground truth up to a permutation and scaling (Theorem 3.3.(ii)). Let $\hat{\mathbf{Z}} \in \mathbb{R}^n$ denote the learned latent representations and $\mathbf{Z} \in \mathbb{R}^n$ the ground truth latent variables. First, we compute the Pearson correlation matrix $\mathrm{Corr}$ between the learned and true variables. Since the learned variables are identifiable only up to permutation, we compute the MCC using the permutation $\pi \in \mathrm{perm}([n])$ that maximizes the average absolute correlation:

$$\mathrm{MCC} = \frac{1}{n} \max_{\pi \in \mathrm{perm}([n])} \sum_{i=1}^{n} \left| \mathrm{Corr}_{i,\pi(i)} \right|.$$

The correlation matrix under the optimal permutation is denoted as $\mathrm{Corr}_\pi^{n \times n}$, calculated with data from all domains stacked together. This metric ranges from 0 to 1, with 1 indicating perfect recovery.

**ii) Domain-wise MCC (d-MCC).** To assess performance in a multi-domain setting, we extend MCC to a domain-wise metric, *d-MCC*. Suppose we have $T$ domains, indexed by $\tau = 1, \ldots, T$, each with its own set of samples $\mathbf{X}^\tau$ and latent variables $\mathbf{Z}^\tau$. For each domain, we compute the MCC as described above:

$$\mathrm{MCC}^\tau = \frac{1}{n} \max_{\pi \in \mathrm{perm}([n])} \sum_{i=1}^{n} \left| \mathrm{Corr}_{i,\pi(i)}^\tau \right|.$$

The d-MCC is then obtained by averaging over all domains:

$$\mathrm{d\text{-}MCC} = \frac{1}{T} \sum_{\tau=1}^{T} \mathrm{MCC}^\tau.$$

This metric evaluates the recovery of latent variables up to permutation and scaling for each domain.

**iii) Amari Distance.** We also report the *Amari distance* (Amari et al., 1995), a standard metric in Independent Component Analysis (ICA) algorithms and blind source separation, which measures the deviation of an estimated unmixing matrix $\hat{A}^{-1}$ from the ground truth mixing matrix $A^{-1}$ up to permutation and scaling. Let $P$ and $L$ denote the optimal permutation and scaling matrices. The Amari error is defined as:

$$\mathrm{Amari}(\hat{A}, A) = \frac{1}{2n} \left[ \sum_{i=1}^{n} \frac{\sum_{j=1}^{n} |G_{ij}|}{\max_j |G_{ij}|} - 1 + \sum_{j=1}^{n} \frac{\sum_{i=1}^{n} |G_{ij}|}{\max_i |G_{ij}|} - 1 \right], \quad G = LP\hat{A}^{-1}A,$$

where we use persudo-inverse of $\hat{A}$ if $d > n$. Lower values indicate better recovery, with 0 corresponding to perfect recovery. We use the standard Amari distance based on *absolute values*. We notice that several recent implementations instead use a squared variant of this metric. While related, the two are not numerically comparable due to the different penalization of off-diagonal terms.

### E.2. Adaptation of Baselines to Multi-Domain Datasets

First of all, QNDIAG, PARAFAC-COV, MHA and iVAE can be directly used in multi-domain settings (details below). For ICA, we concatenate multi-domain data as a single dataset with a mixture model, which can ensure the non-Gaussianity is satisfied for ICA.

In particular, QNDIAG is a joint diagonalization method that aims to find a shared transformation to map a set of covariance matrices from multiple domains to a set of diagonal matrices. PARAFAC-COV also requires multiple tensors (one for each domain) to achieve a unique decomposition. MHA focuses on various covariance matrices that change across domains. For iVAE, we use domain labels as the auxiliary variable $U$, and within each domain, compute the covariance and mean of the post-distribution.

**Why PARAFAC only for diagonal case?** Suppose $C^\tau$ is the covariance matrix to decompose. If $G$ is diagonal, then the matrices $C^\tau = A^T D^\tau G D^\tau A$ have a simple symmetric PARAFAC/CP structure. Since each $D^\tau$ is diagonal,

$$C^\tau = \sum_{r=1}^n g_r d_{\tau r}^2 \, a_r a_r^T,$$

where $a_r$ denotes the $r$-th latent factor from $A$. Stacking the matrices $C^\tau$ into a 3-way tensor $\mathcal{C}(:,:,i) = C^\tau$ gives the symmetric CP decomposition

$$\mathcal{C} = \sum_{r=1}^n a_r \circ a_r \circ h_r,$$

with third-mode coefficients $h_r(\tau) = g_r d_{\tau r}^2$. Thus the first two modes share the same factors $A$, and only the weights vary with $\tau$.

If $G$ is non-diagonal, the decomposition is no longer standard PARAFAC. Expanding the product gives

$$C^\tau = \sum_{r,s} g_{rs} \, d_{\tau r} d_{\tau s} \, a_r a_s^T,$$

which introduces cross-interactions between latent components $r$ and $s$, thus the terms are coupled through the matrix $G$. PARAFAC is recovered only in the special case where $G$ is diagonal (or has special low-rank structure).

### E.3. Numerical Experiments

Following the same setting as in Table 2, we report more results with other metrics: MCC in Table 4 and Amari distance Table 5.

Different from d-MCC used in Table 2 which can access the recovery of domain-invariant mixing matrix $A$, MCC instead can also measure the recovery of domain-specific scaling matrix $D^\tau$ by reflecting whether $(\hat{D}^\tau)^{-1} D^\tau$ are the same for all $\tau \in [T]$. In Table 4, since all other baselines does not specifically estimate the domain-specific scaling, the performance is worse than our method for all settings. Comparing Table 2 and Table 4, we can find that for most of the configurations, our method achieves (near-)perfect recovery according to both MCC and d-MCC. For the less-dependent case ($k = 0, 1$), MCC shows slightly lower values, which means the model recovers well inside each domain but slightly worse across domains. For the low-sample-number case (n=200), MCC is relatively lower than others, which empirically shows that a good recovery for all domains highly relies on the recovery of single domains.

*Table 4.* Results for the numerical experiments: Ave. MCC ± Std over 5 random seeds. The bold font indicates which parameters vary across each block of rows, as well as the best performance method.

| $n$ | $d$ | $T$ | $N$ | $k$ | SCMs | MuDo-CoM | FastICA | QNDIAG | PARAFAC | iVAE | MHA |
|---|---|---|---|---|---|---|---|---|---|---|---|
| **5** | $n$ | $2n$ | 5000 | 2 | Gauss | **0.999±.000** | 0.767±.053 | 0.800±.043 | 0.611±.193 | 0.715±.088 | 0.728±.046 |
| **10** | $n$ | $2n$ | 5000 | 2 | Gauss | **0.999±.001** | 0.740±.033 | 0.788±.029 | 0.433±.219 | 0.634±.072 | 0.687±.045 |
| **20** | $n$ | $2n$ | 5000 | 2 | Gauss | **0.985±.021** | 0.732±.021 | 0.794±.016 | 0.274±.072 | 0.606±.083 | 0.581±.019 |
| 10 | **1**$n$ | $2n$ | 5000 | 2 | Gauss | **0.999±.001** | 0.740±.033 | 0.788±.029 | 0.433±.219 | 0.646±.042 | 0.687±.045 |
| 10 | **2**$n$ | $2n$ | 5000 | 2 | Gauss | **0.998±.001** | 0.741±.034 | 0.788±.029 | 0.443±.174 | 0.634±.072 | 0.653±.034 |
| 10 | **4**$n$ | $2n$ | 5000 | 2 | Gauss | **0.980±.037** | 0.741±.034 | 0.788±.029 | 0.382±.231 | 0.716±.034 | 0.632±.034 |
| 10 | **10**$n$ | $2n$ | 5000 | 2 | Gauss | **0.999±.000** | 0.741±.034 | 0.788±.029 | 0.375±.209 | 0.710±.063 | 0.614±.051 |
| 10 | $n$ | **1**$n$ | 5000 | 2 | Gauss | **0.998±.001** | 0.763±.034 | 0.801±.021 | 0.641±.161 | 0.726±.050 | 0.646±.011 |
| 10 | $n$ | **2**$n$ | 5000 | 2 | Gauss | **0.999±.001** | 0.740±.033 | 0.788±.029 | 0.433±.219 | 0.634±.072 | 0.687±.045 |
| 10 | $n$ | **3**$n$ | 5000 | 2 | Gauss | **0.991±.017** | 0.788±.035 | 0.827±.024 | 0.413±.204 | 0.706±.039 | 0.633±.010 |
| 10 | $n$ | $2n$ | **200** | 2 | Gauss | **0.795±.159** | 0.739±.032 | 0.788±.028 | 0.455±.207 | 0.577±.044 | 0.686±.046 |
| 10 | $n$ | $2n$ | **1000** | 2 | Gauss | **0.978±.030** | 0.736±.036 | 0.788±.029 | 0.512±.170 | 0.543±.049 | 0.687±.046 |
| 10 | $n$ | $2n$ | **5000** | 2 | Gauss | **0.999±.001** | 0.740±.033 | 0.788±.029 | 0.433±.219 | 0.634±.072 | 0.687±.045 |
| 10 | $n$ | $2n$ | **10000** | 2 | Gauss | **0.999±.001** | 0.741±.033 | 0.789±.029 | 0.456±.214 | 0.651±.086 | 0.687±.045 |
| 10 | $n$ | $2n$ | 5000 | **0** | Gauss | **0.983±.015** | 0.937±.003 | 0.937±.003 | 0.897±.077 | 0.880±.035 | 0.500±.028 |
| 10 | $n$ | $2n$ | 5000 | **1** | Gauss | **0.984±.029** | 0.820±.013 | 0.849±.010 | 0.535±.138 | 0.709±.068 | 0.607±.035 |
| 10 | $n$ | $2n$ | 5000 | **2** | Gauss | **0.999±.001** | 0.740±.033 | 0.788±.029 | 0.433±.219 | 0.634±.072 | 0.687±.045 |
| 10 | $n$ | $2n$ | 5000 | **3** | Gauss | **0.999±.001** | 0.678±.030 | 0.757±.010 | 0.364±.104 | 0.596±.052 | 0.733±.017 |
| 10 | $n$ | $2n$ | 5000 | 2 | **Gauss** | **0.999±.001** | 0.740±.033 | 0.788±.029 | 0.433±.219 | 0.634±.072 | 0.687±.045 |
| 10 | $n$ | $2n$ | 5000 | 2 | **Exp** | **0.990±.016** | 0.723±.036 | 0.789±.029 | 0.443±.174 | 0.613±.060 | 0.687±.045 |
| 10 | $n$ | $2n$ | 5000 | 2 | **Gumbel** | **0.982±.034** | 0.715±.047 | 0.762±.042 | 0.425±.234 | 0.632±.072 | 0.676±.045 |
| 10 | $n$ | $2n$ | 5000 | 2 | **Nonlinear** | **0.999±.001** | 0.825±.037 | 0.910±.009 | 0.451±.164 | 0.852±.022 | 0.528±.026 |

Both the Amari distance and d-MCC are used to assess the recovery performance of $A$, however, from different perspectives. Amari distance is directly computed on the composition of ground truth $A$ and estimator $\hat{A}$, whereas MCC is computed on the ground truth latent variables $\mathbf{Z}$ and estimator $\hat{\mathbf{Z}}$. As we can see, the Amari distance is more sensitive to the accuracy of recovery, even in cases where d-MCCs are all $\geq 0.99$, we can still distinguish the difference measured by the Amari distance. The measurements from Amari distance and MCC are consistent with each other.

*Table 5.* Results for the numerical experiments: Ave. Amari ± Std over 5 random seeds. The bold font indicates which parameters vary across each block of rows, as well as the best performance method.

| $n$ | $d$ | $T$ | $N$ | $k$ | SCMs | MUDO-COM | FASTICA | QNDIAG | PARAFAC | MHA |
|---|---|---|---|---|---|---|---|---|---|---|
| **5** | $n$ | $2n$ | 5000 | 2 | GAUSS | **0.07±0.02** | 0.86±0.10 | 0.80±0.08 | 1.00±0.18 | 1.43±0.33 |
| **10** | $n$ | $2n$ | 5000 | 2 | GAUSS | **0.10±0.02** | 1.04±0.13 | 0.94±0.11 | 1.80±0.25 | 3.28±0.26 |
| **20** | $n$ | $2n$ | 5000 | 2 | GAUSS | **0.19±0.03** | 1.09±0.11 | 0.94±0.10 | 2.79±0.63 | 6.31±0.16 |
| 10 | **1**$n$ | $2n$ | 5000 | 2 | GAUSS | **0.10±0.02** | 1.04±0.13 | 0.94±0.11 | 1.80±0.25 | 3.28±0.26 |
| 10 | **2**$n$ | $2n$ | 5000 | 2 | GAUSS | **0.11±0.01** | 1.04±0.13 | 0.94±0.14 | 1.77±0.51 | 3.36±0.40 |
| 10 | **4**$n$ | $2n$ | 5000 | 2 | GAUSS | **0.12±0.04** | 1.04±0.13 | 0.94±0.12 | 1.69±0.30 | 3.41±0.31 |
| 10 | **10**$n$ | $2n$ | 5000 | 2 | GAUSS | **0.10±0.01** | 1.04±0.12 | 0.94±0.12 | 1.56±0.20 | 3.25±0.12 |
| 10 | $n$ | **1**$n$ | 5000 | 2 | GAUSS | **0.16±0.02** | 1.03±0.15 | 0.82±0.03 | 1.59±0.22 | 3.39±0.36 |
| 10 | $n$ | **2**$n$ | 5000 | 2 | GAUSS | **0.10±0.02** | 1.04±0.13 | 0.93±0.16 | 1.80±0.25 | 3.28±0.26 |
| 10 | $n$ | **3**$n$ | 5000 | 2 | GAUSS | **0.08±0.01** | 0.93±0.10 | 0.83±0.10 | 1.48±0.30 | 3.33±0.17 |
| 10 | $n$ | $2n$ | **200** | 2 | GAUSS | **0.77±0.20** | 1.04±0.13 | 0.96±0.12 | 1.97±0.14 | 3.28±0.27 |
| 10 | $n$ | $2n$ | **1000** | 2 | GAUSS | **0.33±0.12** | 1.04±0.13 | 0.94±0.12 | 1.71±0.16 | 3.28±0.26 |
| 10 | $n$ | $2n$ | **5000** | 2 | GAUSS | **0.10±0.02** | 1.04±0.13 | 0.94±0.11 | 1.80±0.25 | 3.28±0.26 |
| 10 | $n$ | $2n$ | **10000** | 2 | GAUSS | **0.08±0.02** | 1.04±0.13 | 0.94±0.12 | 1.80±0.29 | 3.28±0.26 |
| 10 | $n$ | $2n$ | 5000 | **0** | GAUSS | 0.23±0.09 | 0.04±0.01 | **0.02±0.00** | 0.09±0.07 | 3.29±0.26 |
| 10 | $n$ | $2n$ | 5000 | **1** | GAUSS | **0.12±0.03** | 0.59±0.06 | 0.51±0.05 | 1.07±0.18 | 3.29±0.26 |
| 10 | $n$ | $2n$ | 5000 | **2** | GAUSS | **0.10±0.02** | 1.04±0.13 | 0.94±0.11 | 1.80±0.25 | 3.28±0.26 |
| 10 | $n$ | $2n$ | 5000 | **3** | GAUSS | **0.13±0.04** | 1.34±0.26 | 1.12±0.09 | 1.91±0.12 | 3.29±0.26 |
| 10 | $n$ | $2n$ | 5000 | 2 | **GAUSS** | **0.10±0.02** | 1.04±0.13 | 0.94±0.11 | 1.80±0.25 | 3.28±0.26 |
| 10 | $n$ | $2n$ | 5000 | 2 | **EXP** | **0.18±0.10** | 1.07±0.13 | 0.94±0.12 | 1.82±0.23 | 3.29±0.26 |
| 10 | $n$ | $2n$ | 5000 | 2 | **GUMBEL** | **0.15±0.08** | 1.08±0.13 | 0.95±0.12 | 1.68±0.26 | 3.29±0.26 |
| 10 | $n$ | $2n$ | 5000 | 2 | **NONLINEAR** | **0.10±0.02** | 0.92±0.32 | 0.37±0.06 | 1.40±0.09 | 3.28±0.26 |

In addition, we also conduct experiments with signed scaling $D^\tau$. As shown in Fig. 3(a), the low MCC but high d-MCC reflects that there is no global identifiability achieved, but inside each domain, the recovery of latent is almost perfect. Fig. 3(b) shows that the Amari distance decreases steadily as $T$ increases, showing that the estimated $\hat{A}$ becomes more accurate with more domains.

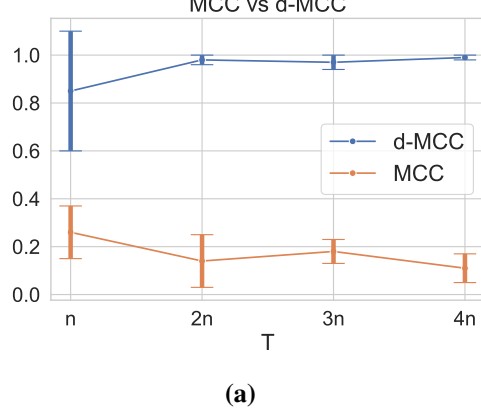

**(a)**

| $T$ | Amari $\downarrow$ |
|---|---|
| $n$ | $0.61 \pm 0.25$ |
| $2n$ | $0.40 \pm 0.16$ |
| $3n$ | $0.38 \pm 0.16$ |
| $4n$ | $0.28 \pm 0.09$ |

**(b)**

*Figure 3.* **Effect of increasing the number of domains** $T$**.** (**a**) Mean $\pm$ standard deviation of MCC and d-MCC over $T \in \{n, 2n, 3n, 4n\}$ across 5 random seeds. (**b**) Corresponding Amari distance, showing improved common mixing matrix recovery as more domains are observed.

# F. Discussion of Assumptions

In this section, we assess the applicability and robustness of our assumptions and discuss to what extent they hold or may fail in practical scenarios.

## F.1. Noiseless Mixing Process

In general, our identifiability theorem can be extended to noisy settings when the noise variance is known. Here, we empirically test whether our model can still identify the latent, even when the noise variance is unknown. We assume the variance of the noise as one more set of parameters to train. The ground truth standard deviation (std) of noise $\sigma$ changes from 0 (no noise) to 1 (heavy noise). In Table 6, we report the estimated std of noise and Amari distance between $\hat{A}$ and ground truth $A$, which shows that empirically our method can recover both the noise variance $\sigma^2$ and the mixing matrix $A$. Setup: $n = d = 10, T = 2n, N = 5000, k = 2$.

*Table 6.* Average Amari destance and $\hat{\sigma}$ over 5 random seeds

| $\sigma$ | 0 | 0.1 | 0.5 | 1 |
|---|---|---|---|---|
| $\hat{\sigma}$ | 0.01 | 0.11 | 0.51 | 1.01 |
| $Amari$ | 0.10 | 0.10 | 0.11 | 0.12 |

## F.2. Linear Mixing Function

We provide some empirical evidence on violations of linear mixing function. We test our method for the nonlinear mixing functions to show the necessity of this assumption. We use an MLP with Leaky-ReLU to model the nonlinear mixing function, where $m$ denotes the number of Leaky-ReLU layers. As expected, the results in Table 7 degrade with more non-linearities, but it is still reasonable with one non-linearity. Setup: $n = d = 5, T = 2n, N = 5000, k = 2$.

*Table 7.* Average $MCC$ over 5 random seeds

| $m$ | 0 | 1 | 2 | 3 |
|---|---|---|---|---|
| $MCC$ | 1.00 | 0.93 | 0.81 | 0.73 |

## F.3. Sensitivity to Cross-Domain Variations

Our theoretical analysis assumes that the mixing matrix and the latent correlation matrix are shared across domains, while domain differences come from component-wise scalings. If the mixing matrix or the latent correlation matrix also varies across domains, this becomes a model misspecification. In such cases, we expect MuDo-CoM to be reasonably robust to moderate deviations, but performance should gradually degrade as the deviation increases.

To assess this empirically, we conduct experiments where

   i.  the mixing matrix is perturbed across domains as $A^\tau = A + \delta_A \Delta_A^\tau$ and

  ii.  the latent correlation matrix varies mildly across domains as $Corr^\tau = Corr + \delta_{Corr} \Delta_{Corr}^\tau$.

The perturbation level is controlled by $\delta_A$ and $\delta_{Corr}$, and $\Delta_A^\tau$ and $\Delta_{Corr}^\tau$ are generated from standard Gauss. As shown in Table 8, the results show that the method can deal with mild misspecifications, but not with larger ones. A full theoretical analysis of identifiability under such misspecification is beyond the scope of the current paper. Setup: $n = d = 10, T = 2n, N = 5000, k = 2$.

*Table 8.* Average $MCC$ over 5 random seeds

| $\delta_A$ | 0 | 0.05 | 0.1 | 0.2 |
|---|---|---|---|---|
| $MCC$ | 1.00 | 0.95 | 0.85 | 0.68 |
| $\delta_{Corr}$ | 0 | 0.05 | 0.1 | 0.2 |
| $MCC$ | 1.00 | 0.98 | 0.90 | 0.52 |

### F.4. Assumption 3.1: Sufficient Domain Variability

In this section, we provide two possible heuristics to test sufficient domain variability empirically:

- Compute the set of domain-wise covariance matrices of observations: If the covariance matrices are nearly identical or linearly dependent, this indicates insufficient domain variability.

- Running random seeds to see the consistency: Similar to what we did in fMRI data, we can run the algorithm multiple times with different random seeds and compare the recovered components across runs. When domains are sufficiently diverse, the estimated unmixing is stable and consistent (up to standard ambiguities such as sign and permutation). In contrast, insufficient variability typically leads to instability or non-unique solutions. This can be quantified, for example, by computing the mean correlation coefficient (MCC) between runs; low MCC indicates insufficient domain diversity.

## G. Discussion

**The gap between theory and practice.**   Our theoretical analysis relies on several simplifying assumptions that create a gap between theory and practical applications. First, we assume a linear mixing model, whereas real-world data may have nonlinear entanglement from latent to observations. Second, we consider a noiseless mixing process, i.e., there is no randomness encoded in the observations beyond the latent components, whereas practical measurements typically contain observation noise. Third, the theoretical guarantees Theorem 4.1 are derived in the asymptotic sense with an infinite number of samples and do not directly characterize sample efficiency, as well as assuming obtaining the global optimal, and do not consider optimization in our study scope. Despite these idealizations, our empirical results, especially on real datasets, indicate that the proposed method is robust to moderate deviations from these assumptions, suggesting that the theory captures the core structure needed for recovery even in realistic settings.

**Sufficient number of domains.**   Our identifiability results require a sufficient number of domains with diverse scaling patterns to ensure the uniqueness of the estimation (Theorem 3.2, Theorem 3.3, Theorem 4.1) . This requirement comes from the need to identify the shared mixing structure $A$ using second-order statistics alone. While the theoretical lower bound on the number of domains can be large in the worst case, it should be interpreted as a *sufficient condition* rather than a tight practical requirement. In experiments (Section 6), we observe successful recovery with substantially fewer domains, where (near-)perfect recovery can be achieved with $T = n$ only. This is likely due to additional structure in the dataset and the continuous nature of the scaling parameters. We believe incorporating a specific and realistic structure into the proposed framework could be an interesting direction to explore.

**How to learn the sign of scaling?**   Our analysis (Theorem 3.3.(i)) shows that domain-specific scalings $D^\tau$ are identifiable only up to sign, i.e., there is a domain-dependent sign indeterminacy in the estimated latent variables. This is because second-order statistics are invariant to sign flips. In most ICA literature, resolving the sign of the scaling would require additional assumptions or information, such as constraints on the sign of domain scalings as we already explored in Theorem 3.3.(ii). Some alternative potential options include higher-order moments or weak supervision. Exploring principled ways to eliminate the sign interminancy remains an interesting direction for future work. Beyond thoeratical identification, in practice, this ambiguity can be resolved with a simple post-processing step by aligning the sign of the sample mean of latent dimensions.

**Identifying the latent graphical structure from statistical dependency**   By explicitly allowing statistical dependence among latent variables, our framework enables recovery not only of latent representations but also of their dependency structure (correlation matrix). Our model can therefore be connected with the work in the causality literature, where the estimated latent covariance matrix can be interpreted as encoding a latent graphical structure. While our work focuses on identifiability and recovery of latent variables, extending the framework to explicitly learn the graphs and thereby identify the causal relationships among latent components is a useful direction for future research.

**Moment matching v.s. likelihood-based methods.**   Our approach is based on moment matching, specifically second-order statistics, in contrast to likelihood-based methods that require explicit modeling of latent distributions. The key advantage of this choice is that estimation does not require assuming a specific form of distribution, and therefore obtain a simple, scalable optimization objective. (As an side benefit, covariance matching is quite robust against the model producing data

that has deficient rank, as is the case here.) However, likelihood-based approaches can be more statistically efficient when the model is correctly specified, as they exploit all statistical information rather than just second-order statistics. In terms of sample efficiency, as shown in Table 2, our method performs worse with relatively small sample sizes. This is expected, since our method relies solely on covariance estimates, which can be noisy for small datasets.

