# OpenReview forum: "Identifying dependent components from multi-domain linear mixtures"
_ICML.cc/2026/Conference — ICML 2026 regular_

### Official Review · Reviewer_1Qww · 2026-03-02

**Soundness:** 3
**Presentation:** 3
**Significance:** 3
**Originality:** 3
**Overall Recommendation:** 5
**Confidence:** 4

**Summary:**

In this paper, MuDo-COM, a method for the separation of dependent sources from linear mixtures in a multi-domain setting is proposed. Additionally, the authors provides a theoretically sufficient condition for source identification.

The proposed algorithm can be seen as a generalization of the classical non-stationarity-based second order methods (Pham&Cardoso, 2001) allowing correlated sources.

The work is mainly motivated by the source localization problem in EEG studies where the authors assume that sources are shared among subject but they are scaled and share the same mixing matrix.

The method is experimentally evaluated  on synthetically generated datasets and one real EEG dataset; and compared with a subset of ICA methods (Independent Component Analysis) (FastICA, QNDIAG, PARAFAC, iVAE) and one Dependent Component Analysis (DCA) algorithm (MHA).

**Compliance With Llm Reviewing Policy:**

Affirmed.

**Final Justification:**

The authors have provided additional analysis and addressed my concerns correctly.

**Key Questions For Authors:**

1-	Is this loss function in eq. (3) convex with respect to A, {D^\tau}, and G? Can be the proposed algorithm got stuck in local minima? Please, provide some theoretical analysis.

2-	Can you provide some analysis on the stability of source extraction taking into account that matrix (A^T)A could be ill-conditioned?

3-	Please, describe how previous methods were applied to the multi-domain setting since they were designed for a single-domain setting.

**Limitations:**

Yes

**Strengths And Weaknesses:**

Strengths:

-	The new method approaches the Dependent Component Analysis (DCA) problem, for which there are only a few previous works.

-	The paper is technically sound, providing a clear mathematical analysis of identifiability of potentially dependent sources.

-	The new proposed algorithm could be applied to several problems in different applications besides the source localization in EEG studies.

Weaknesses:

-	The method is compared with several ICA methods (Independent Component Analysis) (FastICA, QNDIAG, PARAFAC, iVAE) but only one Dependent Component Analysis (DCA) algorithm (MHA). Personally, I don’t find very useful the comparison with ICA algorithms when source are correlated.

-	The paper includes a large description of Related Work, including classical linear and non-linear ICA, tensor decompositions, causal representation learning, however the paper ignores previous works addressing  explicitly DCA, which is more relevant to this work. See, for example:

[1] Francis R. Bach and Michael I. Jordan. 2002. Tree-dependent component analysis. In Proceedings of the Eighteenth conference on Uncertainty in artificial intelligence (UAI'02). Morgan Kaufmann Publishers Inc., San Francisco, CA, USA, 36–44.

[2] Caiafa, C.F. On the conditions for valid objective functions in blind separation of independent and dependent sources. EURASIP J. Adv. Signal Process. 2012, 255 (2012).

[3] Erdogan A: A family of bounded component analysis algorithms. In IEEE International Conference on Acoustics, Speech and Signal Processing, 2012, ICASSP 2012.

-	The paper does not analyze the convexity of the loss function in eq. (3). Is this loss functionconvex with respect to A, {D^\tau}, and G? It is very important to identify if the chosen optimization algorithm based on gradients could be stuck in local minima.

-	Eq. (5) shows how to compute the latent variables once matrices {D^\tau} and A are estimated. However, it is noted that it involves computing the inverse of matrix (A^T)A, which could be a very unstable operation if matrix (A^T)A is ill-conditioned. The paper should include an analysis of stability of source extraction.

-	The proposed algorithm MuDo-COM is compared with other algorithms that were designed for the single-domain setting. It is not clear how the previous methods were applied to the multi-domain setting.

-	While ICA algorithms are usually applied to EEG source detection, the obtained results on the EEG dataset are not validated since there is no a ground-truth available.

---

> ### Author Rebuttal · Authors · 2026-03-30
>
> Thank you for your careful review, it will surely improve our paper. We address your concerns below.
> ## Adding related works ##
>
> We thank the reviewer for this suggestion, we will include a discussion of these related areas in our revised version. In general these approaches have different sets of assumptions, especially about the latent variable distributions, while we do not make any parametric assumptions there (but then require a linear mixing function and scaling in different domains). In particular:
> - NMF and simplex structured matrix factorization assume that the data is non-negative, but we do not make this assumption;
> - Bounded components analysis, e.g., (Erdogan A. 2012) assumes that the components have bounded distributions. The support of the latents lies inside a geometric shape, another assumption we do not make;
> - Tree dependent component analysis (Bach and Jordan, 2002) assumes that the dependencies between latents follow a tree structure, while we allow for arbitrary patterns;
> - Other methods for blind source separation for dependent and independent sources, e.g. (Caiafa 2012), assume linear dependencies between the latent variables (which means one variable can be written as a linear combination of one other), while we allow for arbitrary dependencies.
> ## (Non)-convexity of the loss function ##
>
> Our loss function is not convex with respect to $\hat{A}$, $\hat{D^\tau}$, and $\hat{\Lambda}$ and $\hat{B}$, so it can in principle get stuck in local minima. On the other hand, this is also true for almost all loss functions in this research field. We provide a simple example to show the nonconvexity:
>
> **Example:**
>  Let $dim(X)=dim(Z)=1$, domain number $T=1$, observed covariance $Cov(X):=C>0$.  In this case $loss=(C-(AD\Lambda)^2)^2$. Note that here $C, A, D, \Lambda$ are all scalars. If we fix nonzero $D, \Lambda$, then we can regard the loss as a function of $A$, $Loss(A)=(D\Lambda)^4A^4-2C(D\Lambda)^2A^2+C^2$. Taking the second derivative at $A=0$, we get $-4C(D\Lambda)^2<0$. Thus the loss is not convex.
>
> This matches our choice to use multiple random initializations to escape bad local minima (3 reinitializations used in Tab2), and keep the best run (lowest loss).
> ## Analysis of the stability of source extraction ##
> From a theoretical point of view, to ensure the identifiability of latent, $(A^T)A$ must be invertible. The reason is that in identifiability theory, a common and often necessary assumption is an injective mixing function, because if two distinct latent values $z_1$ and $z_2$ get mapped to the same observation $x$, then generally it is impossible to distinguish them from $x$ only. In our setup, this is equivalent to a full-column-rank A, which implies a full-rank $(A^T)A$. Then, the stability of source extraction has already been sufficiently tested by computing MCC, which needs to use $ (\hat{A}^T)\hat{A}$ to compute $\hat{Z}$, in Tab2.
>
> For completeness, we also conducted experiments with a non-full-column-rank $A$. In this case, identifiability cannot be achieved anymore for the reason we gave above. Therefore, instead of computing MCC, we report the statistics of the absolute value of the determinant of $ (\hat{A}^T)\hat{A}$ for 50 random seeds and show that the step in Eq. (5) is still stable.
>
> Setup: n=d=10, T=2n, N=5000, k=2.
>
> |mean|min|max|
> |---|---|---|
> |118.18|5.73e-5|2262.07|
> ## Application of baselines to multi-domain settings ##
>
> We want to emphasize that QNDIAG, PARAFAC-COV, MHA and iVAE can be directly used in multi-domain settings (details below). For ICA, we concatenate multi-domain data as a single dataset with a mixture model, which can ensure the non-Gaussianity is satisfied for ICA.
>
> In particular, QNDIAG is a joint diagonalization method that aims to find a shared transformation to map a set of covariance matrices from multiple domains to a set of diagonal matrices. PARAFAC-COV also requires multiple tensors (one for each domain) to achieve a unique decomposition. MHA focuses on various covariance matrices that change across domains. For iVAE, we use domain labels as the auxiliary variable $U$, and within each domain, compute the covariance and mean of the post-distribution.

---

> > ### Author Rebuttal · Reviewer_1Qww · 2026-04-02
> >
> > I would like to thank the authors for addressing my concerns. The provided additional analysis should be included in the revision of the paper. I have modified my score accordingly.

---

### Official Review · Reviewer_iUe9 · 2026-03-11

**Soundness:** 2
**Presentation:** 3
**Significance:** 2
**Originality:** 2
**Overall Recommendation:** 4
**Confidence:** 4

**Summary:**

The article offers a blind source separation approach for dependent sources. It makes the following assumptions:

1- Availability of multi domain data with the same linear mixing matrix
2- Variability of the scalings of the sources in different domains. There is a precise identifiability condition about these scalings.
3- Correlation matrix of sources in all domains are identical.


Under these conditions, the measurement correlation matrix of each domain can be written in terms of unknown scalings, unkonwn source correlation matrix and unknown mixing.

This approach proposes a measurement covariance matching metric which is sum of the frobenius norm squares of
observed mixture covariances and product of the unknown parameters.

The authors provide an identifiability condition and related theorem regarding the optimization of this metric.

**Compliance With Llm Reviewing Policy:**

Affirmed.

**Final Justification:**

Based on the rebuttal explanations of the authors, I agree with potential applicability to fmri problem. However, the application scope seems limited beyond that.

**Key Questions For Authors:**

1. How would proposed appraoch compare to PARAFAC2, in terms of structure and performance?

2. How sensitive is the algorithm to variations in mixing and correlation matrix from domain to domain.

**Limitations:**

The authors discuss some limitations in Section F.

**Strengths And Weaknesses:**

STRENGTHS

 The proposed objective is actually easily constructable given the existing methods on second order statistics based blind approaches. However, I have not seen exact objective in other sources. I have not also seen anything similar to the identifiability condition on domain gains.


WEAKNESSES
1. I think a closely related approach is PARAFAC2 and that is somehow not discussed and compared in the article.
2. Main issue is having the exact same mixing matrix and the same source correlation matrix for diffent domains. I can not think about a real life setting that is close to this assumption. I do not really think that the multi-subject/session FMRI setting is really a fit to this assumption. In connection, I do not really see how to interpret the results of application to FMRI data. As we do not know the ground truth, it is hard to obtain a conclusive result about the algorithm's success about this experiment.
3. Based on its content and the depth of novelty, the article appears to be more suitable for a signal processing conference/publication.
4. There are various dependent source separation methods exploiting other assumptions leading to identifiability  that are not mentioned in the article, such as  nonnegative matrix factorization, bounded component analysis, simplex structured matrix factorization, and other structured matrix factorization methods.

---

> ### Author Rebuttal · Authors · 2026-03-30
>
> Thank you for your review and suggestions. We address your concern one by one below.
> ## Relation to PARAFAC2 ##
>
> The key distinction between PARAFAC2 and the PARAFAC model used as our baseline are their structural flexibility: PARAFAC2 is designed to handle tensors with one mode varying in size across slices (e.g., differing numbers of rows), whereas PARAFAC assumes consistent dimensions across all modes.
>
> In our setup, each tensor corresponds to a covariance matrix of size $d*d$, i.e., the dimension of observation $X$, which is identical across all domains. As a result, in our setting PARAFAC2 would not add useful flexibility and PARAFAC is sufficient.
> ## Adding related works ##
>
> We thank the reviewer for this suggestion, we will include a discussion of these related areas in our revised version. In general these approaches have different sets of assumptions, especially about the latent variable distributions, while we do not make any parametric assumptions there (but then require a linear mixing function and scaling in different domains). In particular:
> - NMF and simplex structured matrix factorization assume that the data is non-negative, but we do not make this assumption;
> - Bounded components analysis, e.g., (Erdogan A. 2012) assumes that the components have bounded distributions. The support of the latents lies inside a geometric shape, another assumption we do not make;
> - Tree dependent component analysis (Bach and Jordan, 2002) assumes that the dependencies between latents follow a tree structure, while we allow for arbitrary patterns;
> - Other methods for blind source separation for dependent and independent sources, e.g. (Caiafa 2012), assume linear dependencies between the latent variables (which means one variable can be written as a linear combination of one other), while we allow for arbitrary dependencies.
> ## fMRI assumptions ##
>
> The stability of A over subjects (considered as domains) in fMRI is due to the fact that this models largely anatomically defined regions and networks in the brain, which are known to be quite stable over individuals as confirmed for fMRI by Damoiseaux et al (2006).
>
> The idea that the functional connectivities (correlations) between the regions are similar over subjects, is a more speculative hypothesis, and we cannot find a clear justification in the literature at this moment.
>
> Damoiseaux, J. S., Rombouts, S. A., Barkhof, F., Scheltens, P., Stam, C. J., Smith, S. M., & Beckmann, C. F. (2006). Consistent resting-state networks across healthy subjects. Proceedings of the national academy of sciences, 103(37), 13848-13853.
>
> > how to interpret the results of application to FMRI data (...) as we do not know the ground truth
>
> This is indeed a problem with all related work on fMRI. Historically, when ICA was first applied on fMRI, the results were just visually validated as plausible. After ICA had been performed by a great number of authors, the consensus emerged that it finds something meaningful. Later, it was found that the networks found by ICA correlate with a number of cognitive factors, experimental or pathological conditions, and the like. This has been a long process. We can only hope that in future neuroscience publications we, or others, can find the analysis useful. At the moment it is more like a proof-of-concept of our theory.
> ## Sensitivity to cross-domain variations ##
> Our theoretical analysis assumes that the mixing matrix and the latent correlation matrix are shared across domains, while domain differences come from component-wise scalings. If the mixing matrix or the latent correlation matrix also varies across domains, this becomes a model misspecification. In such cases, we expect MuDo-CoM to be reasonably robust to moderate deviations, but performance should gradually degrade as the deviation increases.
>
> To assess this empirically, we add experiments where
>
> (i) the mixing matrix is perturbed across domains as
>  $A^{\tau}=A+\delta_A \Delta^{\tau}_A$,
>
> (ii) the latent correlation matrix varies mildly across domains as $Corr^{\tau}=Corr+\delta_{Corr} \Delta^{\tau}_{Corr}$.
>
> The perturbation level is controlled by $\delta_A$ and $\delta_{Corr} $, and $ \Delta_A^{\tau},\ \Delta_{Corr}^{\tau} $  are generated from standard Gauss. As you can see, the results show that the method can deal with mild misspecifications, but not with larger ones. A full theoretical analysis of identifiability under such misspecification is beyond the scope of the current paper.
>
> Setup: n=d=10, T=2n, N=5000, k=2.
>
> **Average MCC over 5 random seeds**
> |$\delta_A$|0|0.05|0.1|0.2|
> |---|---|---|---|---|
> |MCC|1.00|0.95| 0.85 | 0.68 |
>
> |$\delta_{Corr}$|0|0.05|0.1|0.2|
> |---|---|---|---|---|
> |MCC|1.00| 0.98 | 0.90 | 0.52  |

---

> > ### Author Rebuttal · Reviewer_iUe9 · 2026-04-03
> >
> > I would like to thank authors for their clarifications. The identical linear mixing  and correlation assumptions seems hard to satisfy beyond fmri application. However, based on the authors' clarifications I increased my score.

---

### Official Review · Reviewer_YQuY · 2026-03-13

**Soundness:** 3
**Presentation:** 3
**Significance:** 1
**Originality:** 2
**Overall Recommendation:** 3
**Confidence:** 4

**Summary:**

The paper investigates how to identify dependent latent components from multi-domain linear mixtures, where different domains vary through domain-specific scalings while the latent component distribution remains unchanged. Under sufficient domain variability, the authors show that the latent variables and mixing matrix are identifiable from second-order statistics, and developed a practical estimation method. Empirical studies on synthetic and fMRI data are provided.

**Compliance With Llm Reviewing Policy:**

Affirmed.

**Final Justification:**

While my minor concerns regarding the connection to multi-domain CRL and the writing part have been addressed, my primary concerns about the restrictive nature of the assumptions remain unresolved. In particular, the assumptions of linear mixing, invariance of the latent variable distributions, and invariance of their dependencies are each restrictive on its own. Taken together, they are especially restrictive.

For example, it is highly restrictive to assume that the overall relationships among latent variables remain the same while only the scaling change. While the authors obtain stronger results with such assumptions, these assumptions might not be reasonable in practice:
- Although the authors suggest that these assumptions may be reasonable for fMRI data, the justifications are not fully convincing and difficult to be addressed in further responses, given that researchers have been developing nonlinear ICA approaches and applying them to analyze fMRI data.
- Moreover, prior work (e.g., Zhang et al. (2024)) showed that it is possible to obtain meaningful, albeit weaker, results under much less restrictive assumptions.

To acknowledge the authors' efforts in the rebuttal, I am increasing my score to Weak Reject.

**Key Questions For Authors:**

For main comments, see the "Strengths And Weaknesses" section.

Other comments on writing:
- Page 1 says "This typically requires latent variation across domains using techniques such as interventions ... However, in many environments, it might not be possible to perform interventions or actions". It might not be fair to say that those works require "performing interventions or actions". Interventions can be viewed as a form of distribution shifts, and those works are just making assumptions on the type of distribution shifts; for instance, von Kugelgen et al. (2023) and Zhang et al. (2023) require single-node paired interventions (no action is really needed as long as one knows that the latent distribution changes in such a way), while Ng et al. (2025) and [1] do not need such assumption and only require sufficient variability for the distributions, similar to the type of distribution shift required in Assumption 1.
- Regarding the statement "We introduce very general identifiability results for linearly mixed latent-variable models in multi-domain settings. We do not assume independence, specific forms of mixing, specific distributions, or specific forms of dependencies" in introduction, this claim is somewhat overstated and should be phrased more carefully. The identifiability result is not "very general" as it relies on the strong assumption that distribution shifts occur only through scalings of the latent variables. Moreover, the statement that the paper does not assume "specific forms of mixing" is misleading, since the model assumes **linear mixing**, which is itself a strong assumption.
- Similarly, the statement "In spite of decades of research, there is still no model that would allow for estimating the components if they have arbitrary dependencies with arbitrary linear mixing, not to mention in the fundamental i.i.d. case" is also incorrect and misleading; see [1].

References:

[1] B. Varıcı, E. Acarturk, K. Shanmugam, and A. Tajer. Linear causal representation learning from unknown multi-node interventions. in NeurIPS, 2024.

**Limitations:**

Yes

**Strengths And Weaknesses:**

Strengths:
- The paper is well written and easy to follow. The notations are well defined and the assumptions are clearly stated.
- The theoretical results appear to be sound and rigorous.

Weaknesses regarding assumptions:
- The mixing function is assumed to be linear, which is a strong assumption.
- Only the scalings of the latent variables distribution are allowed to change across domains. In other words, the distribution of latent variables and the dependencies among them are assumed to be invariant. This is another strong and restrictive assumption.
- The paper used fMRI as motivating example, but it is unclear whether there is rigorous justification on why fMRI follows the two assumptions stated above. If there is such justification, I would encourage the authors to include references and a detailed discussion. Otherwise, the assumptions might be too strong to be applicable in practice.

Other weaknesses:
- Several statements in the paper appear to be overstated and would benefit from more careful phrasing (more elaboration in the "Key Questions For Authors" section).
- The work is based on sufficient variability of the scaling (a specific form of distribution shift), which essentially correspond to multi-node soft interventions in the setting of causal representation learning. There are already several existing works that can handle them (see [1] for linear mixing and also Zhang et al. (2024); Ng et al. (2025) for nonparametric mixing). This limits the novelty of the work. The only improvement, as far as I understand, is that the works I mentioned here have rather loose indeterminacy (the latent variables can only be identified up to permutation and entanglement with ancestors or certain neighbors), while the proposed work can identify the latent variables up to trivial indeterminacy (permutation and scaling). This however comes with a cost of strong assumptions, as mentioned above.
    - This connection with existing works should also be discussed in the paper.

---

> ### Author Rebuttal · Authors · 2026-03-30
>
> Thank you for your feedback and for pointing out some ambiguity in our writing. We address your concerns below and in the revised version.
>
> ## Linear mixing function ##
>
> Assuming a linear mixing function is a common assumption in ICA and CRL literature, and in general, without this assumption and the scaling of latent variables in different domains, we cannot prove identifiability in our setting, in which we have arbitrary distributions of latents with arbitrary dependences, unpaired data, and only a change in scaling, which can be seen as a “global” soft intervention on all variables (as we discuss below).
>
> We test our method for the **nonlinear** mixing functions to show the necessity of this assumption. We use an MLP with Leaky-ReLU to model the nonlinear mixing function, where $m$ denotes the number of Leaky-ReLU layers. As expected, the results degrade with more non-linearities, but it is still reasonable with one non-linearity.
>
> Setup: n=d=5, T=2n, N=5000, k=2.
>
> **Average MCC over 5 random seeds**
> |m|0|1|2|3|
> |---|---|---|---|---|
> |MCC|1.00|0.93|0.81|0.73|
> ## fMRI assumptions ##
>
> Linearity in fMRI is a widely-held assumption, starting from linear regression models in the 1990s, and in resting-state studies, linear ICA is a standard method of analysis; early justification can be found in Buxton et al (2004) and Damoiseaux et al (2006). In any case, doing deep learning would not be feasible with the modest numbers of data points even in the largest data bases.
>
> The levels of activations of such regions are quite individual, especially in resting state. This is analyzed in detail by Lee et al (2023), which justifies the change of scalings in the different domains/subjects.
>
> Buxton RB, Uludağ K, Dubowitz DJ, Liu TT (2004). Modeling the hemodynamic response to brain activation. Neuroimage. 23 Suppl 1:S220-33.
>
> Damoiseaux, J. S., Rombouts, S. A., Barkhof, F., Scheltens, P., Stam, C. J., Smith, S. M., & Beckmann, C. F. (2006). Consistent resting-state networks across healthy subjects. Proceedings of the national academy of sciences, 103(37), 13848-13853.
>
> Lee, S., Bijsterbosch, J. D., ... & Douaud, G. (2023). Amplitudes of resting-state functional networks–investigation into their correlates and biophysical properties. NeuroImage, 265, 119779.
> ## Comparison to interventional CRL ##
>
> First, we wanted to clarify a subtle distinction with CRL: in our work, we do not use any causal interpretation of the latent variables nor make any assumption on why they might be dependent. This can be due to causal relations, but they might also be dependent because of latent confounding or selection bias. So while our method can be applied to CRL, which typically assumes that the variables have a causal interpretation and the dependences are due to causal relations, in our setting interventions are not necessary or even potentially well-defined (if we consider that all variables might be dependent due to latent confounding).
>
> Since this is a subtle distinction, we will now consider the CRL setting. In this case, one can indeed see scaling as a type of soft intervention, but we wanted to point out that this is a **“global” soft intervention**, in the sense that it affects all variables simultaneously, and not a simple multi-node intervention that could potentially have different subsets of variables intervened upon.
>
> Moreover, we disagree that providing **element-wise identifiability vs identifiability up to ancestors/intimate neighbours** is a minor improvement, since it enables interpretability and manipulability of the single variables. We think it is then reasonable that for stronger results we require stronger assumptions, and all of these methods provide different trade-offs.
>
> We also wanted to mention that we cited all these works in the paper, with the exception of Varıcı et al. (2024), for which we cite a follow-up work, but we will also include it explicitly. As you mentioned, in the soft intervention case Varıcı et al. (2024) can only identify the variables up to their ancestors. Moreover, the intervention targets should be sufficiently diverse (the binary intervention vectors should be linearly independent), which means their method cannot be applied in our setting, where all the variables change simultaneously (which induces a singular matrix with all 1 entries that breaks Assump.1 in Varıcı et al. (2024)).
> ## Writing ##
>
> In the introduction, we will rephrase our claims and clarify that indeed we do assume a specific form of mixing, linearity, and clarify that we mean in general identifying dependent components is not possible in a purely observational (non-temporal) setting from a single domain.  We will also rephrase and expand the relation to the interventional CRL work on Pg. 1. We also think that modelling changes in distribution as interventions and actions does not require that the agent itself performs the intervention/action, so we will rephrase the sentence on “performing” actions/interventions.

---

> > ### Author Rebuttal · Reviewer_YQuY · 2026-04-04
> >
> > I appreciate the rebuttal. My concerns have been partially resolved. I will take into account the rebuttal in my final assessment.

---

> > > ### Author Response · Authors · 2026-04-06
> > >
> > > Dear reviewer,
> > >
> > > Thanks for acknowledging our rebuttal. Unfortunately, without knowing precisely which of your concerns are not addressed yet or only partially addressed, we cannot really provide a response. Moreover, we are only allowed this last reply, so we cannot really follow up in case you do have some specific questions. So we hope that this is taken into account for the final decision.

---

### Official Review · Reviewer_hqNC · 2026-03-13

**Soundness:** 4
**Presentation:** 3
**Significance:** 2
**Originality:** 3
**Overall Recommendation:** 4
**Confidence:** 3

**Summary:**

This paper studies recovery of linear latent variables in a setting where the latent factors may be dependent, unlike standard ICA which assumes independence. Its main idea is to leverage multiple domains—such as different subjects, sensors, laboratories, or experimental conditions—where the same latent variables are observed through a common linear mixing matrix, while each domain introduces its own diagonal rescaling.

**Compliance With Llm Reviewing Policy:**

Affirmed.

**Final Justification:**

The rebuttal addressed most of my concerns (strict assumption, sign ambiguity etc), so I decided to increase my score.

**Key Questions For Authors:**

1. How robust is MuDo-CoM to noise, finite samples, and mild model mismatch beyond the idealized linear noiseless theory?

2. In practice, how can users tell whether they have enough sufficiently diverse domains for identifiability, and how problematic is the remaining sign ambiguity?

**Limitations:**

1. The theory is idealized. It assumes a linear, noiseless mixing model with asymptotic guarantees, so robustness to realistic noise, nonlinear mixing, and finite-sample optimization is not well characterized.

2. Practical applicability depends on strong multi-domain conditions. Identifiability requires enough domains with sufficiently diverse scaling patterns, and the recovered domain scalings remain identifiable only up to sign without extra assumptions.

**Strengths And Weaknesses:**

Strengths

1. Strong and clean theory: The paper proves identifiability of dependent latent components from second-order statistics in a multi-domain linear setting, which meaningfully extends classical ICA/joint diagonalization.

2. Solid empirical support: It validates the method on both simulations and a real fMRI dataset, showing the approach is not purely theoretical.

Weaknesses

1. Idealized theory: The guarantees rely on a linear, noiseless model and asymptotic analysis, leaving a gap to realistic noisy/nonlinear settings.

2. Scope is somewhat narrow: Identifiability needs enough domains with diverse scalings, and the paper still leaves issues like sign ambiguity unresolved without extra assumptions.

---

> ### Author Rebuttal · Authors · 2026-03-30
>
> Thanks for your review, we respond to your concerns one by one below.
> ## Theory based on linear, noiseless setting with asymptotic guarantees ##
>
> While we agree that our theoretical results rely on common assumptions like linearity and noiselessness, and provide only asymptotic guarantees, as is common in identifiability literature, in some cases our method can also be empirically applied beyond these settings. We provide some results for the noisy and the non-linear setting in the rebuttal and plan to integrate it in the revised paper.
>
> First, all the empirical results in this paper obviously use finite samples and in Tab 2 in the main paper, we investigated the performance of our method with different numbers of samples $N \in \{200, 1000, 5000, 10k\}$. The results show that the method performs reliably even at moderate sample sizes, with performance improving consistently as $N$ increases.
>
> In general, our identifiability theorem can be extended to **noisy** settings when the variance of the noise is given. Here we empirically test whether our model can still identify the latent, even with an **unknown noise variance**. We assume the variance of the noise as one more set of parameters to train. The ground truth std of noise $\sigma$ changes from 0 (no noise) to 1 (heavy noise). Below, we report the estimated std of noise and Amari distance between $\hat{A}$ and ground truth $A$, which shows that empirically our method can recover both the noise variance and the mixing matrix $A$.
>
> Setup: n=d=10, T=2n, N=5000, k=2.
>
> **Average amari and $\hat{\sigma}$ over 5 random seeds**
>
> |$\sigma$|0|0.1|0.5|1|
> |---|---|---|---|---|
> |$\hat{\sigma}$|0.01|0.11|0.51|1.01|
> |amari|0.10|0.10|0.11|0.12|
>
> We also test our method for the **nonlinear** mixing functions to show the necessity of this assumption. We use an MLP with Leaky-ReLU to model the nonlinear mixing function, where $m$ denotes the number of Leaky-ReLU layers. As expected, the results degrade with more non-linearities, but it is still reasonable with one non-linearity.
>
> Setup: n=d=5, T=2n, N=5000, k=2.
>
> **Average MCC over 5 random seeds**
> |m|0|1|2|3|
> |---|---|---|---|---|
> |MCC|1.00|0.93|0.81|0.73|
> ## Sufficient domain variability assumption ##
>
> We would like to emphasize that this is not a limitation specific to our method, but rather a **fundamental requirement** for identifiability: without other assumptions like specific distributions of latents, interventions, multi-view settings or, in our case, the variation across domains, the problem is not identifiable even in principle. Similar conditions can also be found in related areas such as ICA, causal representation learning (CRL), and joint diagonalization methods.
>
> Testing sufficient domain variability empirically is an interesting question and it might require a proper formalization, but we provide two possible heuristics:
>
> 1. **Compute the set of domain-wise covariance matrices of observations**: If the covariance matrices are nearly identical or linearly dependent, this indicates insufficient domain variability.
>
> 2. **Running random seeds to see the consistency**: Similar to what we did in fMRI data, we can run the algorithm multiple times with different random seeds and compare the recovered components across runs. When domains are sufficiently diverse, the estimated unmixing is stable and consistent (up to standard ambiguities such as sign and permutation). In contrast, insufficient variability typically leads to instability or non-unique solutions. This can be quantified, for example, by computing the mean correlation coefficient (MCC) between runs; low MCC indicates insufficient domain diversity.
> ## Sign ambiguity ##
>
> In most ICA literature, assuming $scaling>0$ is a common assumption, which means the scaling keeps the sign of each latent dimension. In our work, we go beyond this standard assumption by not imposing a sign constraint in Theorem 3.2, allowing for sign ambiguity in the latent representation. If desired, this ambiguity can be resolved with a simple post-processing step by aligning the sign of the sample mean of latent dimensions.

---

> > ### Author Rebuttal · Reviewer_hqNC · 2026-04-05
> >
> > Thank you for the detailed rebuttal response which addressed most of my concerns.

---

### Decision · Program_Chairs · 2026-04-30

**Decision:**

Accept (regular)

**Comment:**

This paper studies multi-domain linear mixing with dependent latent components and proposes MuDo-CoM, a covariance-matching method that identifies latent variables and mixing functions from second-order statistics under domain-specific component scaling. It would strengthen the paper to further discuss the identification conditions (such as in fMRI settings) and assess their practical plausibility, as well as to include experimental comparisons with nonlinear methods on real-world fMRI data.